# FSMLP: Modelling Channel Dependencies With Simplex Theory Based Multi-Layer Perceptions In Frequency Domain

## Abstract

Time series forecasting (TSF) plays a crucial role in various domains. While effective for temporal modeling, channel-wise MLPs suffer from overfitting in inter-channel dependency learning. In this paper, we analyze this via Rademacher complexity theory, identifying extreme values as key overfitting catalysts. To mitigate this issue, we propose to constrains weights to a standard simplex (Simplex-MLP), enforcing simpler patterns and reducing extreme value overfitting. Theoretically, we demonstrate that Simplex-MLP exhibits reduced susceptibility to overfitting on extreme values and demonstrates enhanced generalization capabilities. Based on the Simplex-MLP layer, we propose a novel **F**requency **S**implex **MLP** (FSMLP) framework for time series forecasting, comprising of two kinds of modules: **S**implex **C**hannel-**W**ise **MLP** (SCWM) and **F**requency **T**emporal **MLP** (FTM). Experiments on seven benchmarks confirm FSMLP's accuracy/efficiency improvements and superior scalability. Additionally, simplex-MLP also enhances existing channel-wise MLP methods, reducing their overfitting and boosting performance. code is available.

## 1 Introduction

Time series forecasting (TSF) is crucial across various fields, including web data analysis (Wu et al., 2021), electricity consumption (Trindade, 2015; Zhang et al., 2024; Lai et al., 2018), and weather forecasting (Liu et al., 2024c;d; Cheng et al., 2024; Lim et al., 2021). Accurate predictions from historical data are essential for decision-making, policy development, and strategic planning. Recent advances in deep learning have significantly enhanced TSF capabilities (Xue et al., 2024; Liu et al., 2023; Oreshkin et al., 2020; Luo & Wang, 2024; Zhou et al., 2023; Yi et al., 2024; Du et al., 2022; Li et al., 2023).

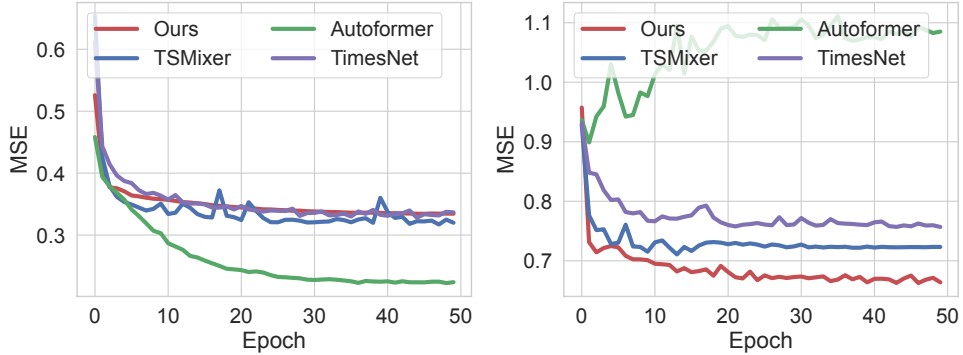

Figure 1: Overfitting on ETTh1: The proposed FSMLP vs. TimesNet, TSMixer, and Autoformer. Among all methods, the proposed FSMLP maintains the lowest validation loss, showing the effectiveness of the Simplex MLP in mitigating overfitting.

Deep learning-based TSF methods can be categorized into channel-independent and channel-mix approaches. Channel-independent methods (Zeng et al., 2023; Xu et al., 2024) focus on capturing

Table 1: In mainstream time series datasets, extreme values are observed, with the majority of values falling within $\sigma$, while a few outliers exceed $3\sigma$. These extreme values significantly impact the performance of MLPs in capturing channel dependencies.

|            | ETTh1  | ETTh2  | ETTm1  | ETTm2  | Traffic | Weather | ECL    |
|------------|--------|--------|--------|--------|---------|---------|--------|
| $\leq \sigma$  | 87.47% | 87.22  | 87.54% | 87.30% | 88.44%  | 86.69%  | 84.38% |
| $\geq 3\sigma$ | 0.35%  | 0.99%  | 0.35%  | 0.99%  | 1.59%   | 0.8%    | 0.39%  |

temporal dependencies by using a shared model across different channels. For example, PatchTST (Nie et al., 2023) segments time series into patches and employs attention mechanisms to capture relationships between these patches, thereby effectively modeling temporal dependencies. Although these methods frequently demonstrate strong performances, their inability to explicitly model inter-channel dependencies through dedicated mechanisms (Liu et al., 2024c; Luo & Wang, 2024; Ilbert et al., 2024) may substantially limit their effectiveness in scenarios where cross-channel interactions encode critical diagnostic information.

Unlike channel-independent methods, channel-mix models excel at capturing inter-channel relationships by explicitly modeling the dependencies among channels. There exist various approaches to model these dependencies, among which one of the simplest and most intuitive methods is the use of MLPs (Multi-Layer Perceptrons). This approach has been employed in early Transformer models (Zhou et al., 2022b; Wu et al., 2021; Liu et al., 2023; Zhou et al., 2021; Ni et al., 2024), such as Fedformer (Zhou et al., 2022b), Autoformer (Wu et al., 2021), Nonstationary-Transformer (Liu et al., 2023), and MLP-Mixer-like models such as TSMixer (Chen et al., 2023). However, despite success in modelling temporal dependencies, these channel-wise MLP models often suffer from overfitting and display sub-optimal performance compared to channel-wise attention methods (Liu et al., 2024c). As shown in Fig. 1, TimesNet, TSMixer, and Autoformer all exhibit signs of overfitting.

In this paper, **we leverage Rademacher complexity theory (Bartlett & Mendelson, 2003) to analyze this phenomenon and find that this overfitting issue may be potentially caused by the presence of extreme values in time series data**, as shown in Table 1. Rademacher complexity measures the ability of a function class to fit random noise, with lower Rademacher complexity indicating a reduced tendency to overfit. The Rademacher complexity $\mathcal{R}_S(\mathcal{H})$ for the hypothesis class $\mathcal{H}$ of the MLP in linear regression is bounded as follows: $\mathcal{R}_S(\mathcal{H}) \leq \frac{B}{m}\sqrt{\sum_{i=1}^{m}\|x^{(i)}\|_2^2}$ where $w \in \mathbb{R}^d$ are the weight parameters of the model, $d$ is the input dimensionality, $m$ is the number of training data points, $x^{(i)} \in \mathbb{R}^d$ represents the $i$-th input data point, $\|x^{(i)}\|_2$ is the $\ell_2$-norm of the $i$-th data point, and $B$ is an upper bound on the norm of the weight vector $w$, typically assumed to be $B = \|w\|_2$ and much greater than 1. The presence of extreme values in time series data can result in a large $B$ when modeling channel dependencies with MLPs, thereby increasing the Rademacher complexity and making these models more prone to overfitting.

To address the overfitting oriented from the extreme values, **we propose a novel operator: Simplex-MLP, inspired by the theory of the Standard $n$-simplex** (Eaves, 1984). A standard $n$-simplex is defined as the set of points in $\mathbb{R}^{n+1}$ that satisfy two conditions: the sum of the coordinates of each point equals 1, and each coordinate is greater than or equal to zero. We apply the $n$-simplex theory to traditional MLP and constrain the weights of MLP to lie within the standard $n$-simplex. We call such a novel MLP as Simplex-MLP and the Rademacher complexity of it can be bounded by $\mathcal{R}_S(\mathcal{H}_\Delta) \leq \frac{1}{m}\sqrt{\sum_{i=1}^{m}\|x^{(i)}\|_2^2}$, much smaller than that of traditional MLPs. This indicates that the Simplex-MLP reduces the influence of redundant noise among channels and thereby improving generalization and reducing overfitting. Also, we empirically demonstrate that Simplex-MLP has better generalization than both L1 and L2 Normalization.

Based on the Simplex-MLP, we propose **F**requency **S**implex **MLP** (FSMLP), an innovative channel-mix framework that synergistically integrates frequency-domain analysis for time series forecasting. The architecture comprises two specialized modules: (1) Simplex Channel-Wise MLP (SCWM) for facilitating cross-channel feature interaction through adaptive frequency filtering, and (2) Frequency Temporal MLP (FTM) employing spectral decomposition for multi-scale temporal pattern extraction.

This dual-domain design enables simultaneous learning of inter-channel correlations in frequency domain and the dynamics in time domain, achieving comprehensive signal representation while maintaining computational efficiency (Ye et al., 2024; Yi et al., 2024; 2023; Li et al., 2024).

In summary, our contributions are as follows: (1) We utilized Rademacher complexity to analyze and identify that the use of MLPs for explicitly modeling inter-channel dependencies leads to overfitting, potentially due to the presence of extreme values in time series dataset. (2) To mitigate overfitting, we introduce a novel Simplex-MLP layer, which constrains the weights of the MLP to lie within a well-defined standard $n$-simplex, making the MLP less prone to overfitting. Besides, based on Simplex-MLP, we proposed a novel framework FSMLP for time series forecasting. (3) Experimentally, we **substantially outperform existing state-of-the-art methods across seven widely-used benchmarks**, achieving performance gains of up to 10.7%, which conclusively demonstrates the effectiveness of the proposed Simplex-MLP layer and FSMLP framework. Moreover, through experiments, we also demonstrate that the proposed Simplex-MLP can also greatly improve other methods.

## 2 RELATED WORK

### 2.1 TIME SERIES FORECASTING

Time series forecasting (TSF) is a fundamental task with applications in web analytics (Wu et al., 2021), energy consumption (Trindade, 2015; Zhang et al., 2024; Lai et al., 2018; Zhou et al., 2022a; Xu et al., 2020), and weather prediction (Liu et al., 2024c;d; Cheng et al., 2024; Lim et al., 2021). Deep learning has significantly advanced TSF by enabling the modeling of complex temporal dependencies and nonlinear patterns. Transformer-based models such as Informer (Zhou et al., 2021) and Autoformer (Wu et al., 2021) address long-range dependencies and scalability challenges, achieving state-of-the-art results (Xue et al., 2024; Liu et al., 2023; Oreshkin et al., 2020; Luo & Wang, 2024).

MLP-based methods are also gaining attention due to their simplicity and efficiency. Models like DLinear (Zeng et al., 2023) and RLinear (Kim et al., 2021) achieve strong performance using only linear layers, by decomposing time series into seasonal/trend components or applying instance normalization.

Beyond the time domain, the frequency domain provides a complementary view by highlighting global and periodic patterns (Xu et al., 2024; Ye et al., 2024; Yi et al., 2024). FreTS (Yi et al., 2023) leverages complex-valued MLPs in the frequency domain to model global dependencies. However, it does not explicitly capture inter-channel or temporal dependencies with MLPs, limiting its modeling capacity despite frequency-domain transformations along both the time and channel axes.

### 2.2 CHANNEL-INDEPENDENT AND CHANNEL-MIX METHODS

Deep learning-based TSF methods can be broadly categorized into channel-independent and channel-mix methods. Channel-independent methods (Das et al., 2023; Zeng et al., 2023; Kim et al., 2021; Nie et al., 2023; Sun & Boning, 2022; Ye et al., 2024), such as DLinear (Zeng et al., 2023) and RLinear (Kim et al., 2021), focus on modeling each time series channel with a shared model. These methods generally exhibit robust performance as the input sequence length increases, but not modeling inter-channel dependencies. for example PatchTST (Nie et al., 2023), segments time series into patches and employs attention mechanisms to capture relationships within these patches, achieving significant success without considering inter-channel correlations.

Channel-mix models (Wang et al., 2025; Liu et al., 2024a; Gu & Dao, 2023; Yang et al., 2024; Liu et al., 2024c; Luo & Wang, 2024), on the other hand, aim to capture dependencies between different channels. It's intuitive to capture channel-dependencies by MLPs. This approach has been employed in early Transformer models (Zhou et al., 2022b; Wu et al., 2021; Liu et al., 2023; Zhou et al., 2021; Oreshkin et al., 2020), such as Fedformer (Zhou et al., 2022b), Autoformer (Wu et al., 2021), Nonstationary-Transformer (Liu et al., 2023), and MLP-Mixer-like models such as TSMixer (Chen et al., 2023). TSMixer (Chen et al., 2023), for instance, models inter-channel dependencies using MLPs in the time domain, while FreTS (Yi et al., 2023) utilizes complex-valued MLPs to capture these dependencies in the frequency domain. Despite these innovations, MLP-based models still

face challenges related to overfitting, similar to early methods. Thus, there is a need to develop an enhanced MLP that efficiently captures channel dependencies while maintaining high performance.

## 2.3 STRUCTURED MLPS

A line of work focuses on enhancing the efficiency and generalization of MLPs through structured matrix designs. Butterfly matrices (Dao et al., 2020) introduce recursive divide-and-conquer structures to accelerate linear transforms, while Monarch (Dao et al., 2022) proposes block-diagonal compositions for hardware-friendly training of large models. BOFT (Liu et al., 2024b) leverages orthogonal butterfly structures to enable parameter-efficient fine-tuning of large-scale vision and language models. These methods typically incorporate low-rank or sparse parameterizations to mitigate overfitting and reduce computational cost. However, they do not address the specific challenge of extreme values in time series forecasting, particularly the overfitting that arises from modeling inter-channel dependencies. In contrast, FSMLP explicitly handles this overlooked problem without relying on low-rank or sparse designs, offering a distinct and effective approach for robust time series forecasting.

## 3 PRELIMINARIES

### 3.1 STANDARD $n$-SIMPLEX

An $n$-simplex (Eaves, 1984) is a generalization of the notion of a triangle or tetrahedron to $n$ dimensions. It is defined as the convex hull of its $n + 1$ vertices in $\mathbb{R}^n$. Specifically, the $n$-simplex is the set of points $\mathbf{w}$ such that $\Delta^n = \left\{ \mathbf{w} \in \mathbb{R}^n \mid \mathbf{w} = \sum_{i=0}^{n} \lambda_i \mathbf{v}_i, \sum_{i=0}^{n} \lambda_i = 1, \lambda_i \geq 0 \right\}$, where $\mathbf{v}_i \in \mathbb{R}^{n+1}$ is the $i$-th vertice of the simplex.

In standard $n$-simplex, each $i$-th vertice $\mathbf{v}_i$ is a standard basis vector. Formally, the standard $n$-simplex $\Delta^n$ is given by $\Delta^n = \left\{ \mathbf{w} \in \mathbb{R}^{n+1} \mid \sum_{i=0}^{n} w_i = 1 \text{ and } w_i \geq 0 \text{ for all } i \right\}$.

This can be visualized as the convex hull of the $n + 1$ standard basis vectors $\mathbf{e}_i$ in $\mathbb{R}^{n+1}$, where $\mathbf{e}_i = (0, \ldots, 0, 1, 0, \ldots, 0)$, for $i = 0, 1, \ldots, n$.

### 3.2 PROBLEM DEFINITION

Let $[X_1, X_2, \cdots, X_T] \in \mathbb{R}^{N \times T}$ stand for the regularly sampled multi-channel time series dataset with $N$ series and $T$ timestamps, where $X_t \in \mathbb{R}^N$ denotes the multi-channel values of $N$ distinct series at timestamp $t$. We consider a time series lookback window of length-$L$ at each timestamp $t$ as the model input, namely $\mathbf{X}_t = [X_{t-L+1}, X_{t-L+2}, \cdots, X_t] \in \mathbb{R}^{N \times L}$; also, we consider a horizon window of length-$\tau$ at timestamp $t$ as the prediction target, denoted as $\mathbf{Y}_t = [X_{t+1}, X_{t+2}, \cdots, X_{t+\tau}] \in \mathbb{R}^{N \times \tau}$. Then the time series forecasting formulation is to use historical observations $\mathbf{X}_t$ to predict future values $\mathbf{Y}_t$. For simplicity, we shorten the model input $\mathbf{X}_t$ as $\mathbf{X} = [X_1, X_2, \cdots, X_L] \in \mathbb{R}^{N \times L}$ and reformulate the prediction target as $\mathbf{Y} = [X_{L+1}, X_{L+2}, \cdots, X_{L+\tau}] \in \mathbb{R}^{N \times \tau}$, in the rest of the paper.

## 4 METHODOLOGY

In this section, we first provide the details of Simplex-MLP and then describe the architecture of the proposed FSMLP.

### 4.1 SIMPLEX-MLP

Traditional MLP is commonly formulated as:

$$X_{\text{out}} = \text{Matmul}(X_{\text{in}}, W) + b, \tag{1}$$

where the weights $W$ are unconstrained. This may lead to large or unbounded values that overfit the data, particularly in high-dimensional settings, as shown in Fig. 1. To address the overfitting problem commonly encountered in traditional linear layers when modeling complex channel dependencies,

we introduce the **Simplex-MLP** layer. The key motivation behind this approach lies in the imposition of geometric constraints on the weight space, specifically confining weights to reside within the **Standard N-Simplex**. This constraint inherently bounds weights within a well-defined geometric region (Eaves, 1984), thereby enabling weight learning in a restricted parameter space while effectively mitigating the risk of overfitting to extreme values.

Theoretically, we demonstrate that such geometric constraint contributes to reducing the Rademacher complexity—a measure of a model's capacity to fit random noise—which crucially enhances generalization guarantees (Bartlett & Mendelson, 2003). Our empirical analysis demonstrates that the Simplex constraint achieves superior generalization performance compared to the other two widely-adopted overfitting mitigation strategies—L1 and L2 normalization. We formulate the Simplex MLP as:

$$X_{\text{out}} = \text{Matmul}(X_{\text{in}}, f_{\text{sim}}(W)) + b, \tag{2}$$

where $f_{\text{sim}}$ is the function to constrain weights $W$ to lie on the Standard N-Simplex. Both the proof and the experimental results in the Appendix verify that based on the proposed Simplex MLP, the model becomes less susceptible to memorizing spurious correlations or noise in the data, leading to significantly reduced overfitting.

The $f_{\text{sim}}$ function can be detailed as $f_{\text{norm}}(f_{\text{trans}}(W))$, where $f_{\text{trans}}$ and $f_{\text{norm}}$ are a normalization and a transformation operators, respectively. The operator $f_{\text{trans}}$ can be realized with each of the three following functions: **Absolute Value Transformation**: For each element $W_{i,j}$ of the weight matrix $W$, we apply the absolute value function: $f_{\text{trans}}^{\text{abs}}(W_{i,j}) = |W_{i,j}|$. **Logarithmic Transformation (Log with Offset)**: First, we take the absolute value of the weights, add a constant (usually 1) to avoid taking the logarithm of zero, and then apply the logarithm: $f_{\text{trans}}^{\log}(W_{i,j}) = \log(|W_{i,j}| + 1)$. **Square Transformation**: Each element $W_{i,j}$ of the weight matrix is squared to ensure positivity and a more concentrated mapping: $f_{\text{trans}}^{\text{square}}(W_{i,j}) = W_{i,j}^2$.

We default set $f_{\text{trans}}$ to the logarithmic transformation $f_{\text{trans}}^{\log}$ because the derivative of the log function is an inverse function, meaning that larger values of the weights result in smaller gradients during optimization. This leads to a reduction in the rate at which the weights grow, making it more difficult for the weights to reach excessively large values. We also evaluate the other two kinds of transformations.

After applying any of these transformations to the weights, we deploy a normalization to the second dimension (the dimension corresponding to the channels) of the weight to ensure that the

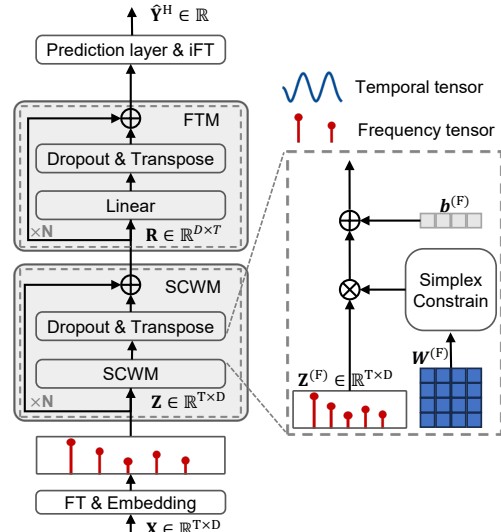

Figure 2: The overall architecture of the proposed FSMLP. We first extract the inter-channel dependencies with our Simplex Channel-Wise MLP then use the Frequency Temporal MLP to extract temporal dependencies.

weights sum to one along this dimension. Specifically, we normalize the weights by dividing each column of the transformed matrix by the sum of the elements in that column. Therefore, we can formulate $f_{\text{sim}}$ as

$$f_{\text{sim}}(W_{i,j}) = f_{\text{norm}}(f_{\text{trans}}(W_{i,j})) = \frac{f_{\text{trans}}(W_{i,j})}{\sum_{j=1}^{N} f_{\text{trans}}(W_{i,j})}$$

Finally, the normalized weight matrix $f_{\text{sim}}(W_{i,j})$ is guaranteed to lie on the Standard N-Simplex, as the weights are positive, sum to one, and respect the structural constraints of the simplex. We provide detailed theoretical analysis in the Appendix.

## 4.2 OVERALL ARCHITECTURE

Based on the Simplex-MLP, we propose FSMLP, a framework composing of two kinds of blocks: Frequency **S**implex **C**hannel-**W**ise **MLP** (**SCWM**) and **F**requency **T**emporal **MLP** (**FTM**). Fig. 2 shows the architecture of the proposed FSMLP, where $N$ SCWM blocks and $N$ FTM blocks are cascaded respectively to gradually capture the complex inter-channel dependencies and frequency temporal relations. As mentioned above, each frequency component corresponds to a specific period in the time domain. By modeling inter-channel dependencies in the frequency domain, the model captures variations across different periods in different channels, which helps mitigate overfitting to noise in time series data, as it focuses on the underlying periodic patterns rather than the noise inherent in the time-domain representation (Yi et al., 2023; Zhou et al., 2022b). We perform Frequency Transformation (FT) to convert the input into the frequency domain, allowing the model to process frequency representations rather than time-domain data. After the linear forecasting head, we apply an inverse Frequency Transformation (iFT) to convert the frequency domain output back to the time domain for the final prediction.

### 4.2.1 SCWM BLOCK

The SCWM block consists of two main steps. First, to extract inter-channel information, we use the proposed Simplex MLP to capture inter-channel dependencies. For the $l$-th SCWM block, we formulate this step as $\mathbf{Z}_{\text{Channel}}^{l} = \sigma(f_{\text{sim}}(\mathbf{Z}_{\text{SCWM}}^{l-1})) + \mathbf{Z}_{\text{SCWM}}^{l-1}$, where $\mathbf{Z}_{\text{SCWM}}^{l-1}$ represents the output from the $l$-1-th SCWM block, and $\sigma$ is the activation function applied to the output.

Next, we extract the temporal information using a simple one-layer MLP as $\mathbf{Z}_{\text{SCWM}}^{l} = \sigma(\text{MLP}(\mathbf{Z}_{\text{Channel}}^{l})) + \mathbf{Z}_{\text{Channel}}^{l}$, where $\mathbf{Z}_{\text{SCWM}}^{l}$ represents the output of the $l$-th block SCWM.

### 4.2.2 FTM BLOCK

Given the output of the $N$-th SCWM block: $\mathbf{Z}_{\text{SCWM}}^{N}$, we apply $N$ cascaded FTM blocks to further capture temporal dependencies in the time series. This process can be expressed as $\mathbf{Z}_{\text{FTM}}^{i} = \sigma(Linear(\mathbf{Z}_{\text{FTM}}^{i-1})) + \mathbf{Z}_{\text{FTM}}^{i-1}$, where $\mathbf{Z}_{\text{FTM}}^{i-1}$ represents the output from the $i-1$ th FTM block. Then, we utilize a linear layer as our forecast head, as $\hat{\mathbf{Y}} = Linear(\mathbf{Z}_{\text{FTM}}^{N})$, where $\hat{\mathbf{Y}}$ is the prediction.

## 4.3 LOSS FUNCTION

To thoroughly leverage the advantages of the frequency and time domains, we propose calculating the loss function specifically in each domain. For the time domain, we use Mean Squared Error (MSE) as the loss function, while for the frequency domain, we utilize Mean Absolute Error (MAE) loss instead of MSE. The reason why we use L1 loss function in frequency domain is that different frequency components often exhibit vastly varying magnitudes, rendering squared loss methods unstable. The overall loss function of our method can be expressed as:

$$\begin{cases} \mathcal{L}_{\text{time}} = \sum_{i=1}^{\tau} \frac{\|\mathbf{Y}_i - F(\mathbf{X})_i\|^2}{\tau}, \\ \mathcal{L}_{\text{fre}} = \sum_{i=1}^{\tau} \frac{\|\mathbf{Y}_i - F(\mathbf{X})_i\|}{\tau}, \\ \mathcal{L}_{\text{total}} = \mathcal{L}_{\text{time}} + \mathcal{L}_{\text{fre}}. \end{cases} \tag{3}$$

## 5 THEORETICAL ANALYSIS

We theoretically analyze why Simplex-MLP outperforms standard MLPs in handling extreme values and resisting overfitting using Rademacher complexity — a measure of how well a model fits random noise, where lower values indicate better generalization. Crucially, since Rademacher complexity depends on weight norms, our analysis follows two key insights: (1) extreme values in training data drastically inflate weight norms in unconstrained models, while (2) Simplex-MLP exhibits strictly lower Rademacher complexity than unconstrained counterparts, explaining its performance.

### 5.1 NORM INFLATION CAUSED BY EXTREME VALUES

In this section, we formally establish through theoretical analysis that extreme values induce weight norm growth in standard MLPs. The following theorem quantifies this relationship:

**Theorem 1** (Weight Norm Growth with Extreme Values). *Consider a linear regression model* $\mathcal{M}$ *with weight matrix* $W \in \mathbb{R}^{d \times q}$, *trained via gradient descent on data* $(X, Y)$. *Let* $(\hat{X}, \hat{Y}) = (X + \Delta X, Y + \Delta Y)$ *contain* $K_x$ *and* $K_y$ *extreme values in* $X$ *and* $Y$, *respectively, with* $|\Delta X_{ij}| \geq \delta_x$, $|\Delta Y_{ij}| \geq \delta_y$. *If* $\|W\|_F \leq \Gamma$ *and* $K_x + K_y \geq \frac{4\Gamma^2}{\min(\delta_x^2, \delta_y^2)\lambda_{\min}^{-2}}$, *where* $\lambda_{\min}$ *is the smallest eigenvalue of* $X^\top X$, *then* $\|\hat{W}\|_F^2 - \|W\|_F^2 > 0$.

This theorem demonstrates that even a limited number of high-magnitude outliers ($K_x + K_y$) in the training data necessarily leads to an increase in the Frobenius norm of the learned weights. This norm expansion directly enhances the model's effective capacity, thereby creating conditions conducive to overfitting. (The proof can be found in the Appendix.)

## 5.2 GENERALIZATION CAPACITY VIA RADEMACHER COMPLEXITY

Our comparative Rademacher complexity analysis reveals that Simplex-MLP achieves strictly better generalization bounds than unconstrained MLPs. For simplicity, we assume a scalar output $y$, though the analysis extends naturally to multi-dimensional settings. The theoretical results demonstrate:

**Unconstrained MLP.** For a linear model with weight norm bounded by $B$, the Rademacher complexity is:

$$\mathcal{R}_S(\mathcal{H}) \leq \frac{B}{m}\sqrt{\sum_{i=1}^m \|x^{(i)}\|_2^2},$$

where $x^{(i)}$ denotes $i$-th data point and $m$ is the total number of training samples.

**Simplex-MLP.** The Simplex-MLP constrains each weight vector $w$ to lie in the standard simplex: $\Delta^n = \{w \in \mathbb{R}^n \mid w_i \geq 0, \sum_{i=1}^n w_i = 1\}$. This constraint on the weights ensures that each component is non-negative and that the sum of the weights is exactly $1$. This constraint enforces that the model cannot assign disproportionately large weights to any feature, preventing it from overfitting to noisy or extreme values in the data.

**Theorem 2** (Rademacher Complexity of Simplex-MLP). *For the constrained hypothesis space* $\mathcal{H}_\Delta = \{f_w(x) = w^\top x \mid w \in \Delta^n\}$, *the complexity satisfies:*

$$\mathcal{R}_S(\mathcal{H}_\Delta) \leq \frac{1}{m}\sqrt{\sum_{i=1}^m \|x^{(i)}\|_2^2}$$

Theorem above reveals that Simplex-MLP enjoys strictly better generalization than standard MLPs in extreme regimes. While conventional MLPs would see their Rademacher complexity bound grow with increasing weight norm $B$, Simplex-MLP's fundamental weight normalization mechanism prevents this growth. This structural advantage yields significantly lower complexity bounds, particularly beneficial for large-scale or ill-conditioned inputs.

# 6 EXPERIMENTS

## 6.1 EXPERIMENT SETTINGS

We evaluate the efficacy of FSMLP on time series forecasting, demonstrating its potential as a foundation model with competitive performance. Our study uses seven widely-adopted multi-channel time series forecasting datasets from diverse domains, including ETTh1, ETTh2, ETTm1, ETTm2, ECL, Traffic, and Weather (Wu et al., 2021), adhering to standardized data-splitting protocols for fairness. We compare FSMLP against a comprehensive set of state-of-the-art baselines. Channel-independent methods include PatchTST (Nie et al., 2023), RLinear and DLinear (Kim et al., 2021; Zeng et al., 2023), SCINet (Liu et al., 2022), and FITS (Xu et al., 2024). Channel-mix methods include Crossformer (Zhang & Yan, 2022), FEDformer (Zhou et al., 2022b), Autoformer (Wu et al., 2021), iTransformer (Liu et al., 2024c), TSMixer (Chen et al., 2023), FreTS (Yi et al., 2023), FiLM (Zhou et al., 2022a) and ARM (Lu et al., 2023). Following established practices (Wu et al.,

Table 2: Full results on the long-term forecasting task with forecast lengths $\tau = 96, 192, 336$ and $720$. The length of history window is set to 96 for all baselines. *Avg* indicates the results averaged over forecasting lengths.

| Models | FSMLP (Ours) | | iTransformer (2024) | | ARM (2024) | | FreTS (2023) | | TSMixer (2023) | | TimesNet (2023) | | Crossformer (2023) | | TiDE (2023) | | DLinear (2023) | | FEDformer (2022) | | PatchTST (2023) | | Autoformer (2021) | | FITS (2024) | |
|---|---|---|---|---|---|---|---|---|---|---|---|---|---|---|---|---|---|---|---|---|---|---|---|---|---|---|
| Metrics | MSE | MAE | MSE | MAE | MSE | MAE | MSE | MAE | MSE | MAE | MSE | MAE | MSE | MAE | MSE | MAE | MSE | MAE | MSE | MAE | MSE | MAE | MSE | MAE | MSE | MAE |
| ETTm1 | **0.365** | **0.382** | 0.407 | 0.410 | 0.379 | 0.385 | 0.407 | 0.415 | 0.529 | 0.513 | 0.400 | 0.406 | 0.558 | 0.532 | 0.419 | 0.419 | 0.404 | 0.407 | 0.440 | 0.451 | 0.387 | 0.400 | 0.596 | 0.517 | 0.422 | 0.421 |
| ETTm2 | **0.265** | **0.311** | 0.288 | 0.332 | 0.280 | 0.324 | 0.335 | 0.379 | 1.030 | 0.753 | 0.297 | 0.329 | 1.633 | 0.782 | 0.358 | 0.404 | 0.344 | 0.396 | 0.302 | 0.348 | 0.281 | 0.347 | 0.326 | 0.366 | 0.289 | 0.351 |
| ETTh1 | **0.416** | **0.425** | 0.454 | 0.497 | 0.443 | 0.448 | 0.488 | 0.474 | 0.623 | 0.585 | 0.458 | 0.450 | 0.628 | 0.574 | 0.541 | 0.507 | 0.462 | 0.458 | 0.441 | 0.457 | 0.469 | 0.454 | 0.476 | 0.477 | 0.442 | 0.430 |
| ETTh2 | **0.350** | **0.384** | 0.383 | 0.407 | 0.369 | 0.395 | 0.550 | 0.515 | 2.025 | 1.194 | 0.413 | 0.426 | 2.136 | 1.130 | 0.611 | 0.550 | 0.558 | 0.516 | 0.430 | 0.447 | 0.387 | 0.407 | 0.478 | 0.483 | 0.377 | 0.398 |
| ECL | **0.159** | **0.252** | 0.178 | 0.270 | 0.189 | 0.276 | 0.209 | 0.297 | 0.233 | 0.340 | 0.214 | 0.307 | 0.182 | 0.279 | 0.251 | 0.344 | 0.225 | 0.319 | 0.229 | 0.339 | 0.205 | 0.290 | 0.228 | 0.339 | 0.224 | 0.298 |
| Traffic | **0.415** | **0.272** | 0.428 | 0.282 | 0.459 | 0.293 | 0.552 | 0.348 | 0.573 | 0.388 | 0.553 | 0.292 | 0.760 | 0.473 | 0.673 | 0.419 | 0.611 | 0.379 | 0.637 | 0.399 | 0.481 | 0.304 | 1.001 | 0.652 | 0.652 | 0.388 |
| Weather | **0.237** | **0.264** | 0.258 | 0.278 | 0.257 | 0.275 | 0.255 | 0.299 | 0.251 | 0.305 | 0.262 | 0.288 | 0.262 | 0.324 | 0.271 | 0.320 | 0.265 | 0.317 | 0.311 | 0.361 | 0.259 | 0.281 | 0.349 | 0.391 | 0.251 | 0.276 |

Table 3: Improvement of Autoformer and TSMixer with Simplex-MLP.

| | ETTh1 | | ETTh2 | | ETTm1 | | ETTm2 | | Traffic | |
|---|---|---|---|---|---|---|---|---|---|---|
| | MSE | MAE | MSE | MAE | MSE | MAE | MSE | MAE | MSE | MAE |
| TSMixer | 0.623 | 0.585 | 2.025 | 1.194 | 0.529 | 0.513 | 1.030 | 0.753 | 0.573 | 0.388 |
| TSMixer(W. Simplex) | **0.553** | **0.530** | **0.589** | **0.534** | **0.442** | **0.459** | **0.366** | **0.406** | **0.525** | **0.347** |
| *Improvement* | 7.00%↑ | 5.50%↑ | 143.60%↑ | 66.00%↑ | 8.70%↑ | 5.40%↑ | 66.40%↑ | 34.70%↑ | 4.80%↑ | 4.10%↑ |
| Autoformer | 0.476 | 0.477 | 0.478 | 0.483 | 0.596 | 0.517 | 0.326 | 0.366 | 1.001 | 0.652 |
| Autoformer(W. Simplex) | **0.434** | **0.462** | **0.439** | **0.453** | **0.570** | **0.511** | **0.319** | **0.361** | **0.631** | **0.388** |
| *Improvement* | 4.20%↑ | 1.50%↑ | 3.90%↑ | 3.00%↑ | 2.60%↑ | 0.60%↑ | 0.70%↑ | 0.50%↑ | 37.00%↑ | 26.40%↑ |

2022), we set the look-back window to 96, use instance normalization, and average results over 10 random seeds. Training was conducted with early stopping and optimized with Adam, using DCT for frequency-domain transformation. FSMLP employs three layers with a hidden dimension of 128, balancing simplicity and performance. More details can be found at Appendix.

## 6.2 MAIN RESULTS

The experimental results demonstrate that FSMLP outperforms recent state-of-the-art models in long-term forecasting tasks across various datasets and forecast lengths ($\tau = 96, 192, 336, 720$). While models like FITS, iTransformer, FreTS, and PatchTST exhibit strengths in specific scenarios, FSMLP offers consistent improvements by effectively addressing varying channel dependency complexities.

In datasets with simpler channel dependencies, such as ETTm1 and ETTm2, FSMLP surpasses both simpler approaches like FITS and complex models such as iTransformer and PatchTST. Unlike iTransformer, which suffers from overfitting due to the high parameter count in its attention mechanisms, FSMLP employs Simplex-MLP to constrain the weight space, enhancing generalization and reducing overfitting.

For datasets with complex channel dependencies, such as ECL and Traffic, FSMLP excels by unifying time and channel dependency modeling, outperforming models like FreTS, FITS, and PatchTST. FreTS and FITS struggle with inter-channel dependency modeling due to their reliance on FFT and linear transformations, while PatchTST, though effective in capturing time dependencies, fails to address intricate inter-channel interactions.

By integrating robust regularization and unified dependency modeling, FSMLP delivers superior forecasting performance and generalization, positioning itself as a versatile solution for both simple and complex datasets.

## 6.3 ABLATION STUDY

We conduct an ablation study to assess the contributions of each component in FSMLP (Table 8). Removing the Simplex-MLP constraint results in a consistent performance drop, confirming its role in regularization and improved generalization. Excluding the Frequency Transformation (FT) significantly harms performance on datasets like ETTh1 and Traffic, highlighting the importance of modeling periodic patterns in the frequency domain. Lastly, removing the frequency loss also leads

Table 4: Comparison of Simplex MLP with other constraints.

| Techniques | ETTh1 | | ETTh2 | | ETTm1 | | ETTm2 | | ECL | | Traffic | | Weather | |
|---|---|---|---|---|---|---|---|---|---|---|---|---|---|---|
| | MSE | MAE | MSE | MAE | MSE | MAE | MSE | MAE | MSE | MAE | MSE | MAE | MSE | MAE |
| Compressed MLP | 0.461 | 0.459 | 0.368 | 0.389 | 0.379 | 0.393 | 0.275 | 0.319 | 0.176 | 0.271 | 0.454 | 0.306 | 0.264 | 0.282 |
| L2 Norm | 0.472 | 0.463 | 0.389 | 0.396 | 0.392 | 0.401 | 0.289 | 0.329 | 0.184 | 0.275 | 0.468 | 0.319 | 0.273 | 0.288 |
| L1 Norm | 0.465 | 0.463 | 0.382 | 0.393 | 0.385 | 0.397 | 0.286 | 0.327 | 0.181 | 0.271 | 0.455 | 0.308 | 0.271 | 0.285 |
| Monarch | 0.433 | 0.428 | 0.377 | 0.399 | 0.405 | 0.400 | 0.285 | 0.323 | 0.187 | 0.279 | 0.521 | 0.351 | 0.262 | 0.283 |
| Simplex MLP | **0.416** | **0.425** | **0.350** | **0.384** | **0.365** | **0.382** | **0.265** | **0.311** | **0.159** | **0.252** | **0.415** | **0.272** | **0.237** | **0.264** |

Table 5: The ablation experimental results. All results are averaged over forecasting lengths. w/o means that removing this component but retaining other components.

| | ETTh1 | | ETTh2 | | ETTm1 | | ETTm2 | | Traffic | | Weather | | ECL | |
|---|---|---|---|---|---|---|---|---|---|---|---|---|---|---|
| | MSE | MAE | MSE | MAE | MSE | MAE | MSE | MAE | MSE | MAE | MSE | MAE | MSE | MAE |
| w/o Simplex-MLP | 0.478 | 0.465 | 0.397 | 0.408 | 0.408 | 0.405 | 0.295 | 0.336 | 0.489 | 0.310 | 0.263 | 0.281 | 0.205 | 0.289 |
| w/o Frequency Transformation | 0.422 | 0.432 | 0.359 | 0.391 | 0.379 | 0.392 | 0.272 | 0.320 | 0.421 | 0.281 | 0.245 | 0.272 | 0.165 | 0.261 |
| w/o Frequncy Loss | 0.420 | 0.431 | 0.355 | 0.386 | 0.368 | 0.388 | 0.269 | 0.316 | 0.416 | 0.276 | 0.241 | 0.269 | 0.163 | 0.258 |
| Ours | **0.416** | **0.425** | **0.350** | **0.384** | **0.365** | **0.382** | **0.265** | **0.311** | **0.415** | **0.272** | **0.237** | **0.264** | **0.159** | **0.252** |

to notable degradation, especially on Weather and ECL, indicating its effectiveness in guiding the model toward relevant spectral features and reducing overfitting.

### 6.4 ANALYSIS OF SIMPLEX-MLP

**Different implementations of Simplex-MLP.** This section compares three implementations of Simplex-MLP. The Logarithm implementation outperforms both the Absolute Value and Square implementations in terms of MSE and MAE across all datasets. The Logarithm implementation excels due to its ability to better capture data dependencies while maintaining numerical stability, leading to more accurate and reliable forecasting. In conclusion, the Logarithm implementation proves to be the most effective for handling complex time series data. Details can be found at Appendix.

**Simplex MLP for other methods** Table 9 demonstrates Simplex-MLP's universal efficacy when integrated into TSMixer and Autoformer, significantly reducing overfitting and boosting generalization. The constrained standard n-simplex weight space enables TSMixer to better capture inter-channel dependencies, particularly evident in ETTh2 performance gains. Consistent error reductions across ETTh1/ETTm1/Traffic datasets confirm its effectiveness in diverse forecasting scenarios. These structured representations enhance model stability while maintaining task adaptability, validating Simplex-MLP as a versatile enhancement for time series architectures.

Comparison With other constraints We evaluate Simplex-MLP against standard L1/L2 regularization and compressed MLPs, where only top-$k$ singular values and vectors are retained during weight reconstruction. As shown in Table 4, Simplex-MLP consistently outperforms all baselines across datasets, demonstrating stronger generalization and forecasting accuracy. While compressed MLP achieves competitive results on ETTh2 and ETTm2, its SVD-based formulation is less effective than the structured constraints of Simplex-MLP in mitigating overfitting and capturing inter-channel dependencies. We also compare with Monarch (Dao et al., 2022), a method employing expressive structured matrices (e.g., block-diagonal transforms) to improve efficiency and stability. Despite its design, Monarch still lags behind our approach, particularly on complex datasets. These results confirm the structural advantage of Simplex-MLP in suppressing extreme values and enhancing the robustness of MLP-based time series models.

## 7 CONCLUSION

In this work, we present FSMLP, a novel framework for addressing inter-channel dependencies and mitigating overfitting in time series forecasting. By constraining model weights to the Standard $n$-Simplex, FSMLP enhances regularization and generalization, while frequency domain transformations improve its ability to capture periodic dependencies. Experimental results show that FSMLP outperforms state-of-the-art methods across multiple benchmarks, demonstrating scalability and robustness in large-scale, long-term forecasting tasks. These findings position FSMLP as an efficient and reliable solution for diverse applications, including energy consumption, web data analysis, and weather prediction.

## ETHICS STATEMENT

This study did not involve any human subjects, animal experiments, or personal data, and therefore did not require ethics approval.

## REPRODUCIBILITY STATEMENT

The anonymized source code and datasets required to reproduce our findings are available at the link provided in the abstract.

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

## A   USE OF LARGE LANGUAGE MODELS

During the preparation of this manuscript, we utilized large language models (LLMs), including ChatGPT, to enhance the clarity, coherence, and overall readability of the text. The models were employed solely for language refinement and formatting assistance, while all conceptual contributions, analyses, and experimental results reported in this work were generated entirely by the authors.

## B   THEORETICAL ANALYSIS

### B.1   NORM INFLATION CAUSED BY EXTREME VALUES

**Theorem 3** (Weight Norm Growth with Extreme Values). *Consider a linear regression model $\mathcal{M}$ with weight matrix $W \in \mathbb{R}^{d \times q}$, trained via gradient descent on data $(X, Y)$. Let $(\hat{X}, \hat{Y}) = (X + \Delta X, Y + \Delta Y)$ contain $K_x$ and $K_y$ extreme values in $X$ and $Y$, respectively, with $|\Delta X_{ij}| \geq \delta_x$, $|\Delta Y_{ij}| \geq \delta_y$. If $\|W\|_F \leq \Gamma$ and*

$$K_x + K_y \geq \frac{4\Gamma^2}{\min(\delta_x^2, \delta_y^2)\lambda_{\min}^{-2}},$$

*where $\lambda_{\min}$ is the smallest eigenvalue of $X^\top X$, then the perturbed solution $\hat{W}$ satisfies*

$$\|\hat{W}\|_F^2 - \|W\|_F^2 > 0.$$

*Proof.* Let $W$ be the minimizer of the loss on clean data:

$$L(W) = \frac{1}{2}\|XW - Y\|_F^2,$$

and $\hat{W}$ the minimizer on the perturbed data:

$$\hat{L}(W) = \frac{1}{2}\|(X + \Delta X)W - (Y + \Delta Y)\|_F^2.$$

By optimality, $W$ and $\hat{W}$ satisfy:

$$W = \arg\min_W L(W), \quad \hat{W} = \arg\min_W \hat{L}(W).$$

Expanding the perturbed loss at $W$:

$$\hat{L}(W) = \frac{1}{2}\|(X + \Delta X)W - (Y + \Delta Y)\|_F^2$$

$$= \frac{1}{2}\|XW - Y + \Delta XW - \Delta Y\|_F^2$$

$$= \frac{1}{2}\|XW - Y\|_F^2 + \langle XW - Y, \Delta XW - \Delta Y \rangle + \frac{1}{2}\|\Delta XW - \Delta Y\|_F^2.$$

Now let us lower bound the final term. Define

$$Z := \Delta XW - \Delta Y.$$

We lower bound its squared norm:

$$\|Z\|_F^2 \geq \frac{(K_x + K_y) \cdot \min(\delta_x^2, \delta_y^2) \cdot \|W\|_F^2}{\lambda_{\min}^{-2}}.$$

Indeed, since $|\Delta X_{ij}| \geq \delta_x$ in $K_x$ positions, and similarly for $Y$, we get:

$$\|\Delta XW\|_F^2 \geq K_x \delta_x^2 \|W\|_F^2, \quad \|\Delta Y\|_F^2 \geq K_y \delta_y^2.$$

Using triangle inequality and independence of $\Delta X$ and $\Delta Y$, we write:

$$\|\Delta XW - \Delta Y\|_F^2 \geq \frac{1}{2}\|\Delta XW\|_F^2 - \|\Delta Y\|_F^2$$

$$\geq \frac{1}{2}(K_x\delta_x^2\|W\|_F^2) - K_y\delta_y^2.$$

Now recall the assumption:

$$K_x + K_y \geq \frac{4\Gamma^2}{\min(\delta_x^2, \delta_y^2)\lambda_{\min}^{-2}}.$$

Let us assume without loss of generality that $\delta = \min(\delta_x, \delta_y)$, so we simplify:

$$\|\Delta XW - \Delta Y\|_F^2 \geq \delta^2 \cdot (K_x + K_y) \cdot \frac{\|W\|_F^2}{2}.$$

Now if we choose $(K_x + K_y)$ such that:

$$\delta^2 \cdot (K_x + K_y) \cdot \frac{1}{2} \geq \frac{2\|W\|_F^2}{\lambda_{\min}^{-2}},$$

then:

$$\|\Delta XW - \Delta Y\|_F^2 \geq \frac{4\|W\|_F^2}{\lambda_{\min}^{-2}}.$$

This gives that:

$$\hat{L}(W) - L(W) \geq \frac{1}{2}\|\Delta XW - \Delta Y\|_F^2 > 0.$$

But $\hat{W}$ minimizes $\hat{L}$, so $\hat{L}(\hat{W}) < \hat{L}(W)$, which can only happen if $\hat{W}$ increases its norm to fit the perturbation.

Hence:

$$\|\hat{W}\|_F^2 > \|W\|_F^2,$$

which completes the proof. $\qquad\square$

### B.2 RADEMACHER COMPLEXITY

In this section, we compare the Rademacher complexity Bartlett & Mendelson (2003) of standard MLP and our proposed Simplex-MLP, specifically in the context of regression tasks. This comparison will shed light on why the Simplex-MLP, with its weight constraints, is less prone to overfitting compared to the standard MLP, particularly in the presence of extreme values or noisy data.

For a standard MLP, the upper bound of Rademacher complexity, which measures the model's capacity to fit random noise, can be expressed as follows:

$$\mathcal{R}_S(\mathcal{H}) \leq \frac{B}{m}\sqrt{\sum_{i=1}^{m}\|x^{(i)}\|_2^2},$$

where $\mathcal{H}$ is the hypothesis class of the MLP. Here, $w \in \mathbb{R}^d$ are the weight parameters of the model, with $d$ representing the dimensionality of the input, and $m$ is the number of data points in the training set. The term $x^{(i)} \in \mathbb{R}^d$ denotes the $i$-th input data point, and $\|x^{(i)}\|_2$ is the $\ell_2$-norm. The constant $B$ is an upper bound on the norm of the weight vector $w$, often assumed to be $B = \|w\|_2$.

This bound indicates that the capacity of a standard MLP is influenced by both the norm of the weight vector $B$ and the sum of the squared $\ell_2$-norms of the input data points. Given that the weight parameters are unconstrained, the model possesses significant flexibility to adapt to the data, including any outliers. Consequently, this adaptability can elevate the risk of overfitting, particularly in the presence of extreme values or noise within the data.

Now, consider the Simplex-MLP, where the weight vector is constrained to lie within the standard $n$-simplex. The standard $n$-simplex $\Delta^n$ is defined as:

$$\Delta^n = \left\{ w \in \mathbb{R}^n \mid w_i \geq 0 \text{ for all } i, \sum_{i=1}^{n} w_i = 1 \right\}$$

This constraint on the weights ensures that each component $w_i$ is non-negative and that the sum of the weights is exactly 1. This constraint enforces that the model cannot assign disproportionately large weights to any feature, preventing it from overfitting to noisy or extreme values in the data.

The upper bound of Rademacher complexity of the Simplex-MLP is given by:

$$\mathcal{R}_S(\mathcal{H}_\Delta) \leq \frac{1}{m} \sqrt{\sum_{i=1}^{m} \|x^{(i)}\|_2^2},$$

where $\mathcal{H}_\Delta$ represents the hypothesis class of the Simplex-MLP, in which the weights are constrained to lie within the standard $n$-simplex $\Delta^n$. The key distinction here is that the weight vector $w \in \Delta^n$ must satisfy $w_i \geq 0$ and $\sum_{i=1}^{n} w_i = 1$. This constraint significantly limits the model's flexibility in assigning disproportionately large weights to any specific feature, thereby potentially reducing the risk of overfitting.

**Theorem 3.** *Let $\Delta^n$ denote the standard $n$-simplex: $\Delta^n = \{w \in \mathbb{R}^n \mid w_i \geq 0, \sum_{i=1}^{n} w_i = 1\}$, $\mathcal{H}_\Delta$ be the hypothesis class with weights in the $n$-simplex: $\mathcal{H}_\Delta = \left\{ f_w(x) = w^\top x \mid w \in \Delta^n \right\}$, where $f_w(x)$ is a linear function of the data point $x$ with weight vector $w \in \Delta^n$. The Rademacher complexity $\mathcal{R}_S(\mathcal{H}_\Delta)$ of the hypothesis class $\mathcal{H}_\Delta$ is bounded by:*

$$\mathcal{R}_S(\mathcal{H}_\Delta) \leq \frac{1}{m} \sqrt{\sum_{i=1}^{m} \|x^{(i)}\|_2^2},$$

*where $x^{(i)}$ are the data points.*

*Proof.* The Rademacher complexity $\mathcal{R}_S(\mathcal{H}_\Delta)$ is given by:

$$\mathcal{R}_S(\mathcal{H}_\Delta) = \mathbb{E}_\sigma \left[ \sup_{w \in \Delta^n} \frac{1}{m} \sum_{i=1}^{m} \sigma_i \left\langle w, x^{(i)} \right\rangle \right]$$

$$= \frac{1}{m} \mathbb{E}_\sigma \left[ \sup_{w \in \Delta^n} \left\langle w, \sum_{i=1}^{m} \sigma_i x^{(i)} \right\rangle \right]$$

$$= \frac{1}{m} \mathbb{E}_\sigma \left[ \sup_{w \in \Delta^n} w^\top v \right],$$

where we denote $v = \sum_{i=1}^{n} \sigma_i x^{(i)}$. Since $w \in \Delta^n$, the supremum is attained when $w$ is aligned with $v$, that is:

$$\sup_{w \in \Delta^n} w^\top v = \|v\|_2$$

Thus, the Rademacher complexity becomes:

$$\mathcal{R}_S(\mathcal{H}_\Delta) = \frac{1}{m} \mathbb{E}_\sigma \left[ \|v\|_2 \right] \leq \frac{1}{m} \sqrt{\mathbb{E}_\sigma \left[ \|v\|_2^2 \right]}$$

which immediately follows from the Jensen's inequality for the convex function $\| \cdot \|_2$. Furthermore, expand $v$ and compute $\mathbb{E}_\sigma \left[ \|v\|_2^2 \right]$:

$$\mathbb{E}_\sigma \left[ \|v\|_2^2 \right] = \mathbb{E}_\sigma \left[ \sum_{i=1}^{m} \|\sigma_i x^{(i)}\|_2^2 + \sum_{i=1}^{m} \sum_{j \neq i} \langle \sigma_i x^{(i)}, \sigma_j x^{(j)} \rangle \right]$$

Table 6: Dataset description.

| Datasets | ETTh1 | ETTh2 | ETTm1 | ETTm2 | Traffic | ECL | Weather |
|---|---|---|---|---|---|---|---|
| Channels | 7 | 7 | 7 | 7 | 862 | 321 | 21 |
| Timesteps | 17,420 | 17,420 | 69,680 | 69,680 | 17,544 | 26,304 | 52,696 |
| Granularity | 1 hour | 1 hour | 5 min | 5 min | 1 hour | 1 hour | 10 min |

Since $\sigma_i$ are independent and $\mathbb{E}[\sigma_i^2] = 1$, the cross terms vanish, and we are left with:

$$\mathbb{E}_\sigma \left[ \|v\|_2^2 \right] = \sum_{i=1}^n \|x^{(i)}\|_2^2$$

Thus, the Rademacher complexity is bounded by:

$$\mathcal{R}_S(\mathcal{H}_\Delta) \leq \frac{1}{m} \sqrt{\sum_{i=1}^m \|x^{(i)}\|_2^2}$$

$\square$

## C  EXPERIMENTS

### C.1  EXPERIMENT SETTINGS

In this section, we evaluate the efficacy of FSMLP on time series forecasting. We show that our FSMLP can serve as a foundation model with competitive performance on these tasks.

**Datasets**  Our study delves into the analysis of seven widely-used real-world multi-channel time series forecasting datasets. These datasets encompass diverse domains, including ECL Transformer Temperature (ETTh1, ETTh2, ETTm1, and ETTm2) (Zhou et al., 2021), ECL, Traffic, and Weather, as utilized by Autoformer (Wu et al., 2021). For fairness in comparison, we adhere to a standardized protocol (Liu et al., 2024c), dividing all forecasting datasets into training, validation, and test sets. Specifically, we employ a ratio of 6:2:2 for the ETT dataset and 7:1:2 for the remaining datasets. Refer to Table 6 for an overview of the characteristics of these datasets.

**Baselines**  We compare FSMLP against a variety of state-of-the-art baselines. Channel-independent methods include PatchTST, RLinear (Kim et al., 2021) and DLinear (Zeng et al., 2023) SCINet (Liu et al., 2022), FITS (Xu et al., 2024). Channel-mix methods include Crossformer (Zhang & Yan, 2022), FEDformer (Zhou et al., 2022b), Autoformer (Wu et al., 2021), iTransformer (Liu et al., 2024c), TSMixer (Chen et al., 2023), FreTS (Yi et al., 2023), and FiLM (Zhou et al., 2022a).

**Inplemental Details**  Similar to the settings in (Wu et al., 2022), we set the look-back window to 96 for all datasets. Additionally, we incorporate the instance normalization block and reverse instance normalization (Kim et al., 2021). All reported results are the averages over 10 random seeds. The baseline models used in this study were carefully reproduced with hyperparameters obtained from the TimesNet repository (Wu et al., 2022), following reproducibility verification. Training was conducted over 100 epochs, with early stopping applied and a patience of 10 epochs, as done in (Xu et al., 2024). For all baseline models, the Adam optimizer (Kingma & Ba, 2017) was used. To simplify the implementation, we use the Discrete Cosine Transform (DCT) as our frequency-domain transformation, as it operates solely on real numbers. Finally, we set the number of layers in our method to 3, with a hidden dimension of 128.

### C.2  MAIN RESULTS

The experimental results show that FSMLP outperforms several recent state-of-the-art models across various datasets in long-term forecasting tasks with forecast lengths of $\tau = 96, 192, 336, 720$.

Table 7: Full results on the long-term forecasting task with forecast lengths $\tau = 96, 192, 336$ and $720$. The length of history window is set to 96 for all baselines. *Avg* indicates the results averaged over forecasting lengths.

| Models | | FSMLP (Ours) | | iTransformer (2024) | | FreTS (2023) | | TSMixer (2023) | | TimesNet (2023) | | Crossformer (2023) | | TiDE (2023) | | DLinear (2023) | | FEDformer (2022) | | PatchTST (2023) | | Autoformer (2021) | | FITS (2024) | |
|---|---|---|---|---|---|---|---|---|---|---|---|---|---|---|---|---|---|---|---|---|---|---|---|---|---|
| Metrics | | MSE | MAE | MSE | MAE | MSE | MAE | MSE | MAE | MSE | MAE | MSE | MAE | MSE | MAE | MSE | MAE | MSE | MAE | MSE | MAE | MSE | MAE | MSE | MAE |
| ETTm1 | 96 | 0.303 | 0.342 | 0.334 | 0.368 | 0.339 | 0.374 | 0.479 | 0.470 | 0.338 | 0.375 | 0.375 | 0.415 | 0.364 | 0.387 | 0.345 | 0.372 | 0.389 | 0.427 | 0.329 | 0.367 | 0.468 | 0.463 | 0.365 | 0.380 |
| | 192 | 0.347 | 0.368 | 0.377 | 0.391 | 0.382 | 0.397 | 0.480 | 0.482 | 0.374 | 0.387 | 0.453 | 0.474 | 0.398 | 0.404 | 0.381 | 0.390 | 0.402 | 0.431 | 0.367 | 0.385 | 0.573 | 0.509 | 0.400 | 0.397 |
| | 336 | 0.378 | 0.391 | 0.426 | 0.420 | 0.421 | 0.426 | 0.541 | 0.525 | 0.410 | 0.411 | 0.548 | 0.526 | 0.428 | 0.425 | 0.414 | 0.414 | 0.438 | 0.451 | 0.399 | 0.410 | 0.596 | 0.527 | 0.431 | 0.418 |
| | 720 | 0.433 | 0.428 | 0.491 | 0.459 | 0.485 | 0.462 | 0.616 | 0.574 | 0.478 | 0.450 | 0.857 | 0.713 | 0.487 | 0.461 | 0.473 | 0.451 | 0.529 | 0.498 | 0.454 | 0.439 | 0.749 | 0.569 | 0.492 | 0.491 |
| | Avg | 0.365 | 0.382 | 0.407 | 0.410 | 0.407 | 0.415 | 0.529 | 0.513 | 0.400 | 0.406 | 0.558 | 0.532 | 0.419 | 0.419 | 0.404 | 0.407 | 0.440 | 0.451 | 0.387 | 0.400 | 0.596 | 0.517 | 0.422 | 0.421 |
| ETTm2 | 96 | 0.166 | 0.247 | 0.180 | 0.264 | 0.190 | 0.282 | 0.250 | 0.366 | 0.185 | 0.264 | 0.267 | 0.349 | 0.207 | 0.305 | 0.195 | 0.294 | 0.194 | 0.284 | 0.175 | 0.259 | 0.240 | 0.319 | 0.186 | 0.269 |
| | 192 | 0.229 | 0.289 | 0.250 | 0.309 | 0.260 | 0.329 | 0.492 | 0.559 | 0.254 | 0.307 | 0.472 | 0.479 | 0.290 | 0.364 | 0.283 | 0.359 | 0.264 | 0.324 | 0.241 | 0.302 | 0.300 | 0.349 | 0.249 | 0.306 |
| | 336 | 0.286 | 0.326 | 0.311 | 0.348 | 0.373 | 0.405 | 0.833 | 0.734 | 0.314 | 0.345 | 0.919 | 0.702 | 0.377 | 0.422 | 0.384 | 0.427 | 0.319 | 0.359 | 0.305 | 0.343 | 0.339 | 0.375 | 0.309 | 0.343 |
| | 720 | 0.380 | 0.383 | 0.412 | 0.407 | 0.517 | 0.499 | 2.543 | 1.352 | 0.408 | 0.403 | 4.874 | 1.601 | 0.558 | 0.524 | 0.516 | 0.502 | 0.430 | 0.424 | 0.402 | 0.400 | 0.423 | 0.421 | 0.410 | 0.398 |
| | Avg | 0.265 | 0.311 | 0.288 | 0.332 | 0.335 | 0.379 | 1.030 | 0.753 | 0.297 | 0.329 | 1.633 | 0.782 | 0.358 | 0.404 | 0.344 | 0.396 | 0.302 | 0.348 | 0.281 | 0.347 | 0.326 | 0.366 | 0.289 | 0.351 |
| ETTh1 | 96 | 0.361 | 0.384 | 0.386 | 0.405 | 0.399 | 0.412 | 0.466 | 0.482 | 0.384 | 0.402 | 0.441 | 0.457 | 0.479 | 0.464 | 0.396 | 0.410 | 0.377 | 0.418 | 0.414 | 0.419 | 0.423 | 0.441 | 0.387 | 0.394 |
| | 192 | 0.405 | 0.419 | 0.441 | 0.436 | 0.453 | 0.443 | 0.597 | 0.567 | 0.436 | 0.429 | 0.521 | 0.503 | 0.525 | 0.492 | 0.449 | 0.444 | 0.421 | 0.445 | 0.460 | 0.445 | 0.498 | 0.485 | 0.436 | 0.422 |
| | 336 | 0.444 | 0.440 | 0.487 | 0.458 | 0.503 | 0.475 | 0.677 | 0.618 | 0.491 | 0.469 | 0.659 | 0.603 | 0.565 | 0.515 | 0.487 | 0.465 | 0.468 | 0.466 | 0.501 | 0.463 | 0.506 | 0.496 | 0.478 | 0.443 |
| | 720 | 0.454 | 0.457 | 0.503 | 0.491 | 0.596 | 0.565 | 0.752 | 0.674 | 0.521 | 0.500 | 0.893 | 0.736 | 0.594 | 0.558 | 0.516 | 0.513 | 0.500 | 0.493 | 0.500 | 0.488 | 0.477 | 0.487 | 0.468 | 0.463 |
| | Avg | 0.416 | 0.425 | 0.454 | 0.497 | 0.488 | 0.474 | 0.623 | 0.585 | 0.458 | 0.450 | 0.628 | 0.574 | 0.541 | 0.507 | 0.462 | 0.458 | 0.441 | 0.457 | 0.469 | 0.454 | 0.476 | 0.477 | 0.442 | 0.430 |
| ETTh2 | 96 | 0.277 | 0.328 | 0.297 | 0.349 | 0.350 | 0.403 | 1.056 | 0.806 | 0.320 | 0.364 | 0.681 | 0.592 | 0.400 | 0.440 | 0.343 | 0.396 | 0.347 | 0.391 | 0.302 | 0.348 | 0.383 | 0.424 | 0.293 | 0.340 |
| | 192 | 0.346 | 0.377 | 0.380 | 0.400 | 0.472 | 0.475 | 2.586 | 1.403 | 0.409 | 0.417 | 1.837 | 1.054 | 0.528 | 0.509 | 0.473 | 0.474 | 0.430 | 0.443 | 0.388 | 0.400 | 0.557 | 0.511 | 0.378 | 0.391 |
| | 336 | 0.385 | 0.408 | 0.428 | 0.432 | 0.564 | 0.528 | 2.407 | 1.348 | 0.449 | 0.451 | 3.000 | 1.472 | 0.643 | 0.571 | 0.603 | 0.546 | 0.469 | 0.475 | 0.426 | 0.433 | 0.470 | 0.481 | 0.418 | 0.425 |
| | 720 | 0.394 | 0.424 | 0.427 | 0.445 | 0.815 | 0.654 | 2.051 | 1.218 | 0.473 | 0.474 | 3.024 | 1.399 | 0.874 | 0.679 | 0.812 | 0.650 | 0.473 | 0.480 | 0.431 | 0.446 | 0.501 | 0.515 | 0.419 | 0.436 |
| | Avg | 0.350 | 0.384 | 0.383 | 0.407 | 0.550 | 0.515 | 2.025 | 1.194 | 0.413 | 0.426 | 2.136 | 1.130 | 0.611 | 0.550 | 0.558 | 0.516 | 0.430 | 0.447 | 0.387 | 0.407 | 0.478 | 0.483 | 0.377 | 0.398 |
| ECL | 96 | 0.133 | 0.226 | 0.148 | 0.239 | 0.189 | 0.277 | 0.204 | 0.308 | 0.171 | 0.273 | 0.148 | 0.248 | 0.237 | 0.329 | 0.210 | 0.302 | 0.200 | 0.315 | 0.181 | 0.270 | 0.199 | 0.315 | 0.205 | 0.280 |
| | 192 | 0.150 | 0.242 | 0.162 | 0.253 | 0.193 | 0.282 | 0.218 | 0.329 | 0.188 | 0.289 | 0.161 | 0.263 | 0.236 | 0.330 | 0.210 | 0.305 | 0.207 | 0.322 | 0.188 | 0.274 | 0.215 | 0.327 | 0.202 | 0.281 |
| | 336 | 0.164 | 0.259 | 0.178 | 0.269 | 0.207 | 0.296 | 0.239 | 0.350 | 0.208 | 0.304 | 0.191 | 0.289 | 0.249 | 0.344 | 0.223 | 0.319 | 0.226 | 0.340 | 0.204 | 0.293 | 0.232 | 0.343 | 0.217 | 0.297 |
| | 720 | 0.187 | 0.280 | 0.225 | 0.317 | 0.245 | 0.332 | 0.272 | 0.373 | 0.289 | 0.363 | 0.226 | 0.314 | 0.284 | 0.373 | 0.258 | 0.350 | 0.282 | 0.379 | 0.246 | 0.324 | 0.268 | 0.371 | 0.261 | 0.332 |
| | Avg | 0.159 | 0.252 | 0.178 | 0.270 | 0.209 | 0.297 | 0.233 | 0.340 | 0.214 | 0.307 | 0.182 | 0.279 | 0.251 | 0.344 | 0.225 | 0.319 | 0.229 | 0.339 | 0.205 | 0.290 | 0.228 | 0.339 | 0.224 | 0.298 |
| Traffic | 96 | 0.379 | 0.254 | 0.395 | 0.268 | 0.528 | 0.341 | 0.531 | 0.358 | 0.518 | 0.269 | 0.805 | 0.493 | 0.697 | 0.429 | 0.577 | 0.362 | 0.609 | 0.385 | 0.462 | 0.295 | 1.451 | 0.744 | 0.686 | 0.405 |
| | 192 | 0.405 | 0.264 | 0.417 | 0.276 | 0.531 | 0.338 | 0.566 | 0.387 | 0.551 | 0.285 | 0.756 | 0.474 | 0.647 | 0.407 | 0.603 | 0.372 | 0.633 | 0.400 | 0.466 | 0.296 | 0.842 | 0.622 | 0.623 | 0.374 |
| | 336 | 0.422 | 0.274 | 0.432 | 0.283 | 0.551 | 0.345 | 0.578 | 0.392 | 0.546 | 0.293 | 0.762 | 0.477 | 0.653 | 0.410 | 0.615 | 0.378 | 0.637 | 0.398 | 0.482 | 0.304 | 0.844 | 0.620 | 0.629 | 0.378 |
| | 720 | 0.453 | 0.294 | 0.467 | 0.302 | 0.598 | 0.367 | 0.617 | 0.415 | 0.597 | 0.323 | 0.719 | 0.449 | 0.694 | 0.429 | 0.649 | 0.403 | 0.668 | 0.415 | 0.514 | 0.322 | 0.867 | 0.624 | 0.668 | 0.396 |
| | Avg | 0.415 | 0.272 | 0.428 | 0.282 | 0.552 | 0.348 | 0.573 | 0.388 | 0.553 | 0.292 | 0.760 | 0.473 | 0.673 | 0.419 | 0.611 | 0.379 | 0.637 | 0.399 | 0.481 | 0.304 | 1.001 | 0.652 | 0.652 | 0.388 |
| Weather | 96 | 0.149 | 0.193 | 0.174 | 0.214 | 0.184 | 0.239 | 0.180 | 0.252 | 0.178 | 0.226 | 0.177 | 0.246 | 0.202 | 0.261 | 0.197 | 0.259 | 0.221 | 0.304 | 0.177 | 0.218 | 0.284 | 0.355 | 0.169 | 0.214 |
| | 192 | 0.201 | 0.241 | 0.221 | 0.254 | 0.223 | 0.275 | 0.218 | 0.287 | 0.227 | 0.266 | 0.227 | 0.297 | 0.242 | 0.298 | 0.236 | 0.294 | 0.275 | 0.345 | 0.225 | 0.259 | 0.313 | 0.371 | 0.216 | 0.255 |
| | 336 | 0.259 | 0.283 | 0.278 | 0.296 | 0.272 | 0.316 | 0.261 | 0.321 | 0.283 | 0.305 | 0.278 | 0.346 | 0.287 | 0.335 | 0.282 | 0.332 | 0.338 | 0.379 | 0.278 | 0.297 | 0.359 | 0.393 | 0.271 | 0.293 |
| | 720 | 0.337 | 0.337 | 0.358 | 0.347 | 0.340 | 0.363 | 0.348 | 0.362 | 0.359 | 0.355 | 0.368 | 0.407 | 0.351 | 0.386 | 0.347 | 0.384 | 0.408 | 0.418 | 0.354 | 0.348 | 0.440 | 0.446 | 0.350 | 0.344 |
| | Avg | 0.237 | 0.264 | 0.258 | 0.278 | 0.255 | 0.299 | 0.251 | 0.305 | 0.262 | 0.288 | 0.262 | 0.324 | 0.271 | 0.320 | 0.265 | 0.317 | 0.311 | 0.361 | 0.259 | 0.281 | 0.349 | 0.391 | 0.251 | 0.276 |

Although models such as FITS, iTransformer, FreTS, PatchTST, and others have shown promise, FSMLP demonstrates significant improvements, especially in datasets with varying complexities in channel dependencies.

For datasets with simpler channel dependencies, such as ETTm1 and ETTm2, FSMLP achieves notable improvements over both simpler models like FITS and more complex ones like iTransformer and PatchTST. For instance, FSMLP achieves an average MSE of 0.365 and MAE of 0.382 on ETTm1, outperforming all existing methods, including iTransformer. iTransformer tends to struggle with overfitting, especially as the forecasting length increases. This issue can be attributed to the large number of parameters in its attention mechanism, which allows the model to overly fit to noise present in long-term dependencies.

In contrast, FSMLP effectively addresses this issue by incorporating Simplex-MLP, which constrains the weight space, reducing overfitting and enhancing generalization. This capability becomes particularly evident in datasets with more complex channel dependencies, such as ECL and Traffic, where FSMLP significantly outperforms other models. For instance, on the Traffic dataset, **FSMLP achieves the best average MSE of** 0.415 **and MAE of** 0.272**, surpassing models like FreTS, FITS, PatchTST, and iTransformer**.

A key limitation of FreTS is its exclusive reliance on FFT to capture both time and channel dependencies. While FFT can capture frequency-domain patterns effectively, it fails to model the complex inter-channel dependencies that are critical in datasets like Traffic and ECL, where channel interactions play a pivotal role. Similarly, FITS utilizes FFT for frequency-domain transformations and employs a single linear layer for time dependencies, but it lacks explicit modeling of channel dependencies, which diminishes its effectiveness on datasets with intricate channel interactions.

PatchTST, though leveraging attention mechanisms to capture time dependencies, falls short in modeling inter-channel dependencies. This limitation makes PatchTST less suitable for datasets like ECL and Traffic, where both time and channel dependencies are essential for accurate forecasting. FSMLP, by contrast, excels in capturing both time and channel dependencies within a unified framework, offering a distinct advantage over PatchTST and the other models.

On the other hand, FSMLP outperforms other state-of-the-art models on both simple and complex datasets by effectively modeling both time and channel dependencies. Its ability to regularize through Simplex-MLP and handle long-term forecasting tasks without overfitting makes it an ideal solution, particularly for complex datasets like Traffic and ECL, where intricate channel dependencies play a crucial role.

### C.3   ABLATION STUDY

We conduct an ablation study to analyze the individual contributions of different components in our proposed method. The results are presented in Table 8, where we systematically remove key components while retaining others to evaluate their impact on the overall performance.

**First**, we examine the performance of our model without the Simplex-MLP constraint. The results show a significant performance drop across all datasets. This highlights the critical role of the Simplex-MLP component in regularizing the model, promoting a more structured representation, and ultimately improving generalization.

**Next**, we remove the Frequency Transformation(FT), a key component that allows our model to operate in the frequency domain. Without FT, our model performs suboptimally on datasets such as ETTh1 and Traffic, with MSE values significantly higher compared to the full model. This demonstrates the importance of capturing periodic patterns in the frequency domain to effectively model channel dependencies and reduce overfitting.

**Finally**, we evaluate the model without the frequency loss term. Similar to the Frequency Transformation ablation, removing frequency loss leads to a degradation in performance, especially for datasets such as Weather and ECL, where the model's ability to generalize is compromised. This confirms that incorporating frequency loss contributes to the model's ability to focus on relevant features and further mitigates overfitting.

Table 8: The ablation experimental results. All results are averaged over forecasting lengths. w/o means that removing this component but retaining other components.

| | ETTh1 | | ETTh2 | | ETTm1 | | ETTm2 | | Traffic | | Weather | | ECL | |
|---|---|---|---|---|---|---|---|---|---|---|---|---|---|---|
| | MSE | MAE | MSE | MAE | MSE | MAE | MSE | MAE | MSE | MAE | MSE | MAE | MSE | MAE |
| w/o Simplex-MLP | 0.478 | 0.465 | 0.397 | 0.408 | 0.408 | 0.405 | 0.295 | 0.336 | 0.489 | 0.310 | 0.263 | 0.281 | 0.205 | 0.289 |
| w/o Frequency Transformation | 0.422 | 0.432 | 0.359 | 0.391 | 0.379 | 0.392 | 0.272 | 0.320 | 0.421 | 0.281 | 0.245 | 0.272 | 0.165 | 0.261 |
| w/o Frequncy Loss | 0.420 | 0.431 | 0.355 | 0.386 | 0.368 | 0.388 | 0.269 | 0.316 | 0.416 | 0.276 | 0.241 | 0.269 | 0.163 | 0.258 |
| Ours | **0.416** | **0.425** | **0.350** | **0.384** | **0.365** | **0.382** | **0.265** | **0.311** | **0.415** | **0.272** | **0.237** | **0.264** | **0.159** | **0.252** |

Table 9: Improvement of Autoformer and TSMixer with Simplex-MLP.

| | ETTh1 | | ETTh2 | | ETTm1 | | ETTm2 | | Traffic | |
|---|---|---|---|---|---|---|---|---|---|---|
| | MSE | MAE | MSE | MAE | MSE | MAE | MSE | MAE | MSE | MAE |
| TSMixer | 0.623 | 0.585 | 2.025 | 1.194 | 0.529 | 0.513 | 1.030 | 0.753 | 0.573 | 0.388 |
| TSMixer(W. Simplex) | **0.553** | **0.530** | **0.589** | **0.534** | **0.442** | **0.459** | **0.366** | **0.406** | **0.525** | **0.347** |
| *Improvement* | 7.00%↑ | 5.50%↑ | 143.60%↑ | 66.00%↑ | 8.70%↑ | 5.40%↑ | 66.40%↑ | 34.70%↑ | 4.80%↑ | 4.10%↑ |
| Autoformer | 0.476 | 0.477 | 0.478 | 0.483 | 0.596 | 0.517 | 0.326 | 0.366 | 1.001 | 0.652 |
| Autoformer(W. Simplex) | **0.434** | **0.462** | **0.439** | **0.453** | **0.570** | **0.511** | **0.319** | **0.361** | **0.631** | **0.388** |
| *Improvement* | 4.20%↑ | 1.50%↑ | 3.90%↑ | 3.00%↑ | 2.60%↑ | 0.60%↑ | 0.70%↑ | 0.50%↑ | 37.00%↑ | 26.40%↑ |

## C.4 EFFICIENCY ANALYSIS

In this section, we analyze the efficiency of FSMLP and baselines, the setting is as the same as the main results.

**Inference.** We evaluate the inference time of our proposed FSMLP method and compare it with several state-of-the-art models, as shown in Table 10. The results, measured per 256 samples across various datasets, demonstrate the efficiency of FSMLP.

Our **FSMLP method achieves the fastest inference times on most datasets**, consistently outperforming the compared models. These results highlight FSMLP's efficiency, making it a suitable choice for real-time applications where low latency is crucial. In contrast, models like Autoformer and TimesNet exhibit significantly higher inference times, particularly on larger datasets. The superior efficiency of FSMLP is attributed to its optimized architecture that effectively captures inter-channel dependencies with lower computational overhead, thereby reducing the overall inference time. This efficiency makes FSMLP not only accurate but also practical for deployment in environments where computational resources and time are limited.

**Training.** As shown in Fig. 3, FSMLP demonstrates both effectiveness and efficiency. **The framework not only achieves high forecasting accuracy but also offers significant computational advantages**. FSMLP requires considerably less memory and delivers faster training times compared to several state-of-the-art models, such as iTransformer, PatchTST, FreTS, TSMixer, AutoFormer, and TimesNet. These advantages make FSMLP a highly practical solution for time series forecasting, especially in resource-constrained environments where both memory usage and training speed are critical factors.

Table 10: Inference time per 256 samples.

| | ETTh1 | ETTh2 | ETTm1 | ETTm2 | Weather | ECL | Traffic |
|---|---|---|---|---|---|---|---|
| Autoformer | 0.219s | 0.217s | 0.220s | 0.215s | 0.659s | 1.271s | 2.836s |
| TimesNet | 0.049s | 0.042s | 0.045s | 0.047s | 0.244s | 2.504s | 2.536s |
| iTransformer | 0.021s | 0.020s | 0.021s | 0.024s | 0.061s | 0.121s | 0.203s |
| PatchTST | 0.018s | 0.019s | 0.025s | 0.028s | 0.134s | 0.207s | 0.368s |
| FreTS | 0.019s | 0.017s | 0.022s | 0.022s | 0.023 | 0.069s | 0.105s |
| TSMixer | 0.019s | 0.018s | 0.023s | 0.021 | 0.021s | 0.081s | 0.127s |
| FSMLP(Ours) | **0.018s** | **0.017**s | **0.022s** | **0.020s** | **0.021s** | **0.064s** | **0.106s** |

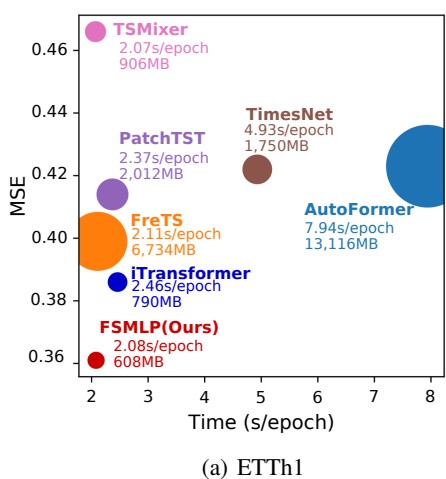 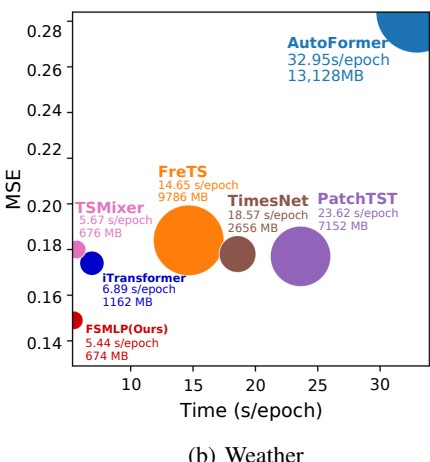

(a) ETTh1      (b) Weather

Figure 3: This is the efficiency comparison of the proposed FSMLP with baselines on ETTh1 dataset with $L = 96$ and $\tau = 96$.

## C.5 ANALYSIS OF SIMPLEX-MLP

### C.5.1 DIFFERENT IMPLEMENTATIONS OF SIMPLEX-MLP

In this section we compare three implementations of Simplex-MLP. As the results shown in Table 11, it is evident that the **Logarithm implementation of Simplex-MLP outperforms both the Absolute Value and Square implementations in terms of MSE and MAE across all datasets.** Specifically, the Logarithm implementation achieves an MSE of 0.416 and an MAE of 0.425 on the ETTh1 dataset, which is superior to the other two implementations. Similar improvements are observed in the ETTh2, ETTm1, ETTm2, and Weather datasets. The superior performance of the Logarithm implementation can be attributed to its ability to better capture the underlying dependencies in the data while maintaining numerical stability. This results in a more accurate and reliable forecasting model, as evidenced by the consistently lower error metrics. In conclusion, the Logarithm implementation of Simplex-MLP demonstrates its effectiveness in handling different time series datasets, making it a preferable choice for capturing complex dependencies and enhancing overall model performance.

### C.5.2 SIMPLEX MLP FOR OTHER METHODS

Table 9 illustrates the improvements achieved by **incorporating Simplex-MLP into TSMixer and Autoformer** across various datasets. The results highlight the significant role of Simplex-MLP in enhancing model performance, particularly by reducing overfitting and improving generalization.

For instance, for TSMixer, the integration of Simplex-MLP yields noticeable performance gains across all datasets. The most significant improvement is observed on the ETTh2 dataset, where the model's ability to capture complex inter-channel dependencies is greatly enhanced. This suggests that Simplex-MLP, by constraining the weight space to the Standard $n$-simplex, promotes a more structured and compact representation of the data, which helps mitigate the overfitting observed in the original TSMixer model. On the other datasets, such as ETTh1, ETTm1, and Traffic, Simplex-MLP also contributes to reducing prediction errors, with consistent gains in both MSE and MAE, highlighting its effectiveness across diverse time series forecasting tasks.

### C.5.3 COMPARISON WITH OTHER CONSTRAINTS

In this section, we compare our Simplex-MLP with other constraints that aim to reduce overfitting, including the L1 Norm, the L2 Norm and compressed MLP. *L1 Norm regularization*, also known as Lasso regularization, adds the absolute value of the weights to the loss function:

$$\text{Loss} = \text{Loss}_{\text{original}} + \lambda \sum_i |w_i|$$

Table 11: Analysis of different kinds of implementations of Simplex-MLP.

| | ETTh1 | | ETTh2 | | ETTm1 | | ETTm2 | | Weather | |
|---|---|---|---|---|---|---|---|---|---|---|
| | MSE | MAE | MSE | MAE | MSE | MAE | MSE | MAE | MSE | MAE |
| Absolute Value | 0.421 | 0.433 | 0.356 | 0.386 | 0.365 | 0.384 | 0.269 | 0.315 | 0.241 | 0.266 |
| Square | 0.419 | 0.428 | 0.353 | 0.387 | 0.367 | 0.385 | 0.267 | 0.313 | 0.239 | 0.267 |
| Log | **0.416** | **0.425** | **0.350** | **0.384** | **0.365** | **0.382** | **0.265** | **0.311** | **0.237** | **0.264** |

*L2 Norm regularization*, also known as Ridge regularization, adds the squared value of the weights to the loss function:

$$\text{Loss} = \text{Loss}_{\text{original}} + \lambda \sum_i w_i^2$$

*Compressed MLP* is a type of MLP that uses singular value decomposition (SVD)

$$W = U\Sigma V^T$$

. Specifically, compressed MLP only stores the largest $k$ singular values and their corresponding vectors. During training, we first reconstruct the weight matrix from these $k$ singular values and vectors before performing the forward pass. In this experiment, we replace the Simplex-MLP in FSMLP with each of the aforementioned constraints and compare their performances with the original Simplex-MLP version.

From Table 4, we observe that Simplex-MLP consistently outperforms L1 Norm, L2 Norm, and Compressed MLP in terms of both MSE and MAE across all datasets. Specifically, on the ETTh1 dataset, Simplex-MLP achieves an MSE of 0.416 and MAE of 0.425, which is **significantly better than Compressed MLP (MSE = 0.461, MAE = 0.459), L2 Norm (MSE = 0.472, MAE = 0.463), and L1 Norm (MSE = 0.465, MAE = 0.463)**. Similarly, for the ECL dataset, Simplex-MLP achieves an MSE of 0.159 and MAE of 0.252, outperforming the other methods by a notable margin.

While Compressed MLP shows competitive results, especially on datasets like ETTh2 and ETTm2, it still falls behind Simplex-MLP in all cases, particularly with respect to the MAE metric. This suggests that the use of singular value decomposition (SVD) in Compressed MLP does not lead to superior performance when compared to the regularization-based methods or our proposed Simplex-MLP.

The Simplex-MLP demonstrates superior performance over traditional regularization methods and compressed MLP techniques in reducing overfitting and improving accuracy. Its ability to constrain the weight space within a well-defined standard simplex enables it to capture inter-channel dependencies effectively while mitigating the impact of extreme values, leading to consistent improvements across various time series forecasting tasks. These results underscore the effectiveness of Simplex-MLP in enhancing the performance of MLP-based models for time series forecasting, making it a valuable addition to the toolkit for handling complex datasets.

## C.6  SCALABILITY

### C.6.1  PARTIAL SAMPLE TRAINING

Table 12 presents the results of the scalability analysis using partial sample training, where both input and prediction lengths are set to 96. The performance of four models—FSMLP, FreTS, TSMixer, and FITS—was evaluated across various datasets, including ETTh1, ETTh2, ETTm1, ETTm2, and Weather. The results indicate that, generally, the models maintain stable performance as the proportion of training data increases. FSMLP demonstrates consistent improvements in performance as more training data is provided, showing enhanced accuracy with additional data. This highlights its scalability and ability to leverage larger training datasets effectively. Moreover, FSMLP consistently shows better performances than the other methods under different training data scale. These findings suggest that FSMLP is highly scalable and well-suited for real-world applications where the volume of training data can vary significantly.

Table 12: Scalability analysis using partial sample training results. Both the input length and the prediction length are 96.

| | | FSMLP | | FreTS | | TSMixer | | FITS | |
|---|---|---|---|---|---|---|---|---|---|
| | | MSE | MAE | MSE | MAE | MSE | MAE | MSE | MAE |
| ETTh1 | 20% | **0.412** | **0.419** | 0.690 | 0.561 | 0.890 | 0.664 | 0.459 | 0.446 |
| | 40% | **0.403** | **0.407** | 0.472 | 0.464 | 0.808 | 0.676 | 0.449 | 0.439 |
| | 60% | **0.389** | **0.399** | 0.442 | 0.449 | 0.598 | 0.567 | 0.438 | 0.436 |
| | 80% | **0.373** | **0.388** | 0.425 | 0.436 | 0.513 | 0.514 | 0.391 | 0.407 |
| | 100% | **0.361** | **0.384** | 0.399 | 0.412 | 0.466 | 0.482 | 0.387 | 0.394 |
| ETTh2 | 20% | **0.305** | **0.353** | 1.11 | 0.73 | 2.420 | 1.17 | 0.306 | 0.359 |
| | 40% | **0.290** | **0.341** | 0.491 | 0.477 | 1.067 | 0.77 | 0.304 | 0.348 |
| | 60% | **0.283** | **0.336** | 0.474 | 0.469 | 1.359 | 0.887 | 0.302 | 0.354 |
| | 80% | **0.282** | **0.336** | 0.498 | 0.475 | 1.198 | 0.824 | 0.299 | 0.352 |
| | 100% | **0.277** | **0.328** | 0.350 | 0.403 | 1.056 | 0.806 | 0.296 | 0.340 |
| ETTm1 | 20% | **0.481** | **0.448** | 1.028 | 0.684 | 0.792 | 0.644 | 0.677 | 0.505 |
| | 40% | **0.495** | **0.442** | 0.689 | 0.546 | 0.794 | 0.632 | 0.563 | 0.479 |
| | 60% | **0.362** | **0.386** | 0.487 | 0.461 | 0.675 | 0.562 | 0.532 | 0.483 |
| | 80% | **0.309** | **0.346** | 0.393 | 0.416 | 0.544 | 0.509 | 0.428 | 0.420 |
| | 100% | **0.303** | **0.342** | 0.339 | 0.374 | 0.479 | 0.470 | 0.365 | 0.380 |
| ETTm2 | 20% | **0.183** | **0.261** | 0.628 | 0.554 | 1.451 | 0.877 | 0.281 | 0.344 |
| | 40% | **0.177** | **0.255** | 0.640 | 0.564 | 0.602 | 0.568 | 0.257 | 0.326 |
| | 60% | **0.173** | **0.252** | 0.428 | 0.455 | 0.464 | 0.503 | 0.217 | 0.315 |
| | 80% | **0.171** | **0.249** | 0.303 | 0.390 | 0.339 | 0.423 | 0.201 | 0.287 |
| | 100% | **0.166** | **0.247** | 0.190 | 0.282 | 0.250 | 0.366 | 0.186 | 0.269 |
| Weather | 20% | **0.161** | **0.204** | 0.268 | 0.296 | 0.384 | 0.407 | 0.189 | 0.233 |
| | 40% | **0.152** | **0.194** | 0.254 | 0.284 | 0.314 | 0.334 | 0.192 | 0.231 |
| | 60% | **0.154** | **0.196** | 0.221 | 0.276 | 0.284 | 0.314 | 0.191 | 0.232 |
| | 80% | **0.154** | **0.199** | 0.205 | 0.257 | 0.244 | 0.286 | 0.191 | 0.231 |
| | 100% | **0.149** | **0.193** | 0.184 | 0.239 | 0.180 | 0.252 | 0.169 | 0.214 |

Table 13: Comparison under longer input length. The results are averaged over prediction lengths.

| | | FSMLP | | FreTS | | FITS | | PatchTST | |
|---|---|---|---|---|---|---|---|---|---|
| | | MSE | MAE | MSE | MAE | MSE | MAE | MSE | MAE |
| ECL | 192 | **0.155** | **0.247** | 0.182 | 0.284 | 0.182 | 0.283 | 0.185 | 0.279 |
| | 336 | **0.153** | **0.246** | 0.175 | 0.279 | 0.168 | 0.272 | 0.162 | 0.263 |
| | 720 | **0149** | **0.240** | 0.178 | 0.281 | 0.163 | 0.265 | 0.159 | 0.261 |
| Traffic | 192 | **0.397** | **0.266** | 0.515 | 0.326 | 0.473 | 0.407 | 0.431 | 0.293 |
| | 336 | **0.389** | **0.261** | 0.483 | 0.317 | 0.431 | 0.295 | 0.401 | 0.278 |
| | 720 | **0.385** | **0.258** | 0.458 | 0.309 | 0.411 | 0.285 | 0.395 | 0.274 |

Table 14: Analysis of different learning rates. All results are averaged over forecasting lengths.

| Learning Rate | ETTh1 | | ETTh2 | | ETTm1 | | ETTm2 | | ECL | |
|---|---|---|---|---|---|---|---|---|---|---|
| | MSE | MAE | MSE | MAE | MSE | MAE | MSE | MAE | MSE | MAE |
| 0.02 | 0.424 | 0.431 | 0.361 | 0.393 | 0.375 | 0.387 | 0.271 | 0.317 | 0.167 | 0.258 |
| 0.01 | **0.416** | **0.425** | **0.350** | **0.384** | **0.365** | **0.382** | **0.265** | **0.311** | **0.159** | **0.252** |
| 0.005 | 0.417 | 0.425 | 0.351 | 0.384 | 0.365 | 0.383 | 0.268 | 0.313 | 0.159 | 0.252 |
| 0.001 | 0.421 | 0.428 | 0.356 | 0.388 | 0.367 | 0.385 | 0.269 | 0.314 | 0.166 | 0.259 |

Table 15: Analysis of different Batch Sizes All results are averaged over forecasting lengths.

| Batch Size | ETTh1 | | ETTh2 | | ETTm1 | | ETTm2 | | ECL | |
|---|---|---|---|---|---|---|---|---|---|---|
| | MSE | MAE | MSE | MAE | MSE | MAE | MSE | MAE | MSE | MAE |
| 512 | 0.424 | 0.431 | 0.357 | 0.389 | 0.372 | 0.389 | 0.273 | 0.319 | 0.167 | 0.258 |
| 256 | **0.416** | **0.425** | **0.350** | **0.384** | **0.365** | **0.382** | **0.265** | **0.311** | **0.159** | **0.252** |
| 128 | 0.419 | 0.427 | 0.353 | 0.386 | 0.367 | 0.384 | 0.267 | 0.313 | 0.159 | 0.252 |
| 64 | 0.420 | 0.427 | 0.352 | 0.386 | 0.368 | 0.384 | 0.268 | 0.314 | 0.162 | 0.254 |

### C.6.2 LARGE DATASET

The Traffic dataset, known for its large scale and complexity, presents a significant challenge for time series forecasting models due to its high dimensionality and the intricate dependencies between different traffic sensors. Our FSMLP method achieves outstanding performance on this dataset, demonstrating its ability to effectively manage and process large-scale datasets. The results show that FSMLP consistently outperforms other models in terms of MSE and MAE, highlighting its robustness and scalability. This performance indicates that FSMLP can generalize well and maintain high accuracy even with the increased data volume and complexity inherent in large-scale datasets, making it a suitable choice for real-world applications that require handling extensive time series data.

### C.6.3 LONGER PREDICTION LENGTHS

Table 16 illustrates the superior performance of FSMLP compared to FreTS, TSMixer, and PatchTST across multiple datasets and increasing prediction lengths. FSMLP's consistent performance, even with extended prediction lengths, highlights its scalability and resilience to overfitting. For datasets like ETTh1 and ETTh2, FSMLP consistently achieves the lowest MSE and MAE values across all tested prediction lengths (960, 1440, and 2160), indicating its capability to handle larger forecasting windows effectively. Similar trends are observed in ETTm1 and ETTm2 datasets, where FSMLP maintains lower errors consistently, showcasing its ability to capture long-term dependencies without overfitting. The Simplex-MLP constraints, which limit weights within the standard $n$-simplex, reduce the influence of redundant noise and enhance generalization. This advantage is further confirmed in the Weather dataset, where FSMLP achieves the lowest error rates, effectively handling complex temporal patterns without overfitting. The combination of frequency domain transformations and Simplex-MLP regularization significantly contributes to FSMLP's robust performance.

### C.6.4 LONGER INPUT LENGTHS

Table 13 provides a comparative analysis of FSMLP, FreTS, FITS, and PatchTST across different input lengths (192, 336, and 720) on the ECL and Traffic datasets. The results, averaged over the prediction lengths, demonstrate that FSMLP consistently achieves the best performance across all input lengths for both datasets. For the ECL dataset, FSMLP shows remarkable stability and superiority, maintaining the lowest error rates compared to other models, which exhibit higher error rates. This highlights FSMLP's ability to effectively utilize longer input sequences without overfitting. In the complex and large-scale Traffic dataset, FSMLP also demonstrates robust performance, consistently outperforming FreTS, FITS, and PatchTST across all input lengths. This underscores

Table 16: Longer prediction length comparison.

| | | FSMLP | | FreTS | | TSMixer | | PatchTST | |
|---|---|---|---|---|---|---|---|---|---|
| | | MSE | MAE | MSE | MAE | MSE | MAE | MSE | MAE |
| ETTh1 | 960 | **0.521** | **0.495** | 0.653 | 0.587 | 0.814 | 0.707 | 0.542 | 0.507 |
| | 1440 | **0.626** | **0.557** | 0.778 | 0.658 | 0.956 | 0.782 | 0.640 | 0.569 |
| | 2160 | **0.813** | **0.648** | 0.969 | 0.752 | 1.114 | 0.855 | 0.842 | 0.662 |
| ETTh2 | 960 | **0.435** | **0.456** | 1.060 | 0.745 | 2.690 | 1.413 | 0.484 | 0.484 |
| | 1440 | **0.532** | **0.516** | 1.467 | 0.880 | 2.884 | 1.484 | 0.551 | 0.525 |
| | 2160 | **0.553** | **0.527** | 1.555 | 0.879 | 3.018 | 1.584 | 0.586 | 0.540 |
| ETTm1 | 960 | **0.478** | **0.448** | 0.540 | 0.504 | 0.684 | 0.605 | 0.494 | 0.458 |
| | 1440 | **0.505** | **0.465** | 0.592 | 0.537 | 0.755 | 0.650 | 0.525 | 0.479 |
| | 2160 | **0.511** | **0.469** | 0.635 | 0.569 | 0.819 | 0.691 | 0.532 | 0.483 |
| ETTm2 | 960 | **0.433** | **0.415** | 0.628 | 0.554 | 1.825 | 1.125 | 0.451 | 0.429 |
| | 1440 | **0.459** | **0.439** | 0.640 | 0.564 | 1.872 | 1.116 | 0.473 | 0.453 |
| | 2160 | **0.449** | **0.443** | 0.734 | 0.622 | 1.891 | 1.136 | 0.478 | 0.461 |
| Weather | 960 | **0.368** | **0.401** | 0.379 | 0.412 | 0.384 | 0.407 | 0.388 | 0.370 |
| | 1440 | **0.381** | **0.397** | 0.394 | 0.420 | 0.409 | 0.436 | 0.419 | 0.389 |
| | 2160 | **0.402** | **0.428** | 0.417 | 0.434 | 0.415 | 0.448 | 0.457 | 0.413 |

Table 17: Comparison of the computational complexity of FSMLP and other baselines. $P$ represents the patch size of PatchTST, $L$ represents the input length, and $N$ represents the number of channels.

| FSMLP | iTransformer | PatchTST | FITS | FreTS | TSMixer |
|---|---|---|---|---|---|
| $O(NL)$ | $O(N^2L)$ | $O\left(\frac{L^2}{P^2}N\right)$ | $O(NL)$ | $O(NL)$ | $O(NL)$ |

FSMLP's scalability and adaptability, effectively capturing long-term dependencies and making it a reliable choice for diverse time series forecasting tasks.

## C.7 COMPLEXITY

Table 17 compares the computational complexity of FSMLP with other state-of-the-art models. FSMLP, FITS, FreTS, and TSMixer all have a linear complexity of $O(NL)$, making them scalable and efficient for large datasets. In contrast, iTransformer has a higher complexity of $O(N^2L)$ due to its use of attention mechanisms to capture inter-channel dependencies, which can lead to higher computational costs and potential overfitting. PatchTST's complexity, $O\left(\frac{L^2}{P^2}N\right)$, reflects its use of patches to model time dependencies. FSMLP's linear complexity, combined with its ability to effectively capture both time and channel dependencies, highlights its efficiency and suitability for large-scale time series forecasting tasks.

## C.8 HYPERPARAMETER SENSITIVITY ANALYSIS

**Learning Rate.** The analysis of different learning rates is summarized in Table 14. As observed, the differences in model performance across the various learning rates are relatively small. The MSE and MAE values show only slight variation between learning rates of 0.02, 0.01, 0.005, and 0.001. This suggests that the model's performance remains fairly stable across the tested learning rates, with no dramatic changes in forecasting accuracy.

**Batch Size.** The analysis of different batch sizes is summarized in Table 15. As shown, the model's performance does not exhibit significant variation across different batch sizes. This suggests that

the model's performance is not highly sensitive to the choice of batch size within the tested range. Although there are small fluctuations, the overall results indicate that different batch sizes provide comparable forecasting accuracy. This indicates that the model can maintain stable performance across a range of batch sizes, offering flexibility in terms of computational efficiency without significantly impacting the predictive outcomes.

