# OpenReview forum: "FSMLP: Modelling Channel Dependencies With Simplex Theory Based Multi-Layer Perceptions In Frequency Domain"
_ICLR.cc/2026/Conference — Submitted to ICLR 2026_

### Official Review · Reviewer_oDSf · 2025-10-28

**Soundness:** 2
**Presentation:** 2
**Contribution:** 2
**Rating:** 4
**Confidence:** 4

**Summary:**

This paper proposes FSMLP, a novel MLP-based architecture for time series forecasting that operates in the frequency domain. The authors begin by providing a theoretical motivation using Rademacher complexity to argue that standard channel-wise MLPs are prone to overfitting, particularly in the presence of extreme values in time series data. To address this, they introduce the "Simplex-MLP," a core component that constrains the weights of the MLP layer to lie within a standard n-simplex, thereby enforcing a form of regularization that encourages simpler patterns and bounds the weight norm.

The proposed FSMLP framework consists of two main modules: (1) a Simplex Channel-Wise MLP (SCWM) that models inter-channel dependencies in the frequency domain, and (2) a Frequency Temporal MLP (FTM) that models temporal dynamics. The authors conduct experiments on seven common time series forecasting benchmarks, demonstrating that FSMLP achieves state-of-the-art performance and is more computationally efficient than many Transformer-based models. They also show that their Simplex-MLP layer can be used to improve the performance of existing models like Autoformer and TSMixer.

**Strengths:**

1. The FSMLP architecture is fully based on MLPs, which makes it computationally efficient in terms of both training and inference time compared to more complex attention-based models. This is a significant practical advantage.
2.  The authors' effort to motivate their approach using Rademacher complexity theory is appreciated. It provides a principled, if not entirely complete, basis for tackling the problem of overfitting in MLP-based models.
3. On the seven benchmark datasets used, FSMLP demonstrates very competitive, often state-of-the-art, performance, showcasing the practical potential of the proposed method in these specific domains.

**Weaknesses:**

1. The primary weakness is that the theoretical analysis does not adequately establish why the simplex constraint is superior to other regularization methods. The analysis shows that the simplex complexity bound is independent of the weight norm, which is interesting, but it doesn't offer a comparative analysis against L1 or L2 norms from a theoretical standpoint. Why is enforcing a sum-to-one and non-negativity constraint the best way to handle overfitting from extreme values, compared to simply penalizing the norm?
2. The comparison could be made more robust by including other recent and strong MLP-based models[1] [2] [3] . The claims of outperforming "existing state-of-the-art methods" would be stronger with a more comprehensive set of contemporary baselines.
3. The experiments are concentrated on datasets from the electricity (ETTh1/2, ETTm1/2, ECL), traffic, and weather domains. This narrow scope makes it difficult to assess the true generalizability of FSMLP. The model's effectiveness on other types of time series data, such as from finance, healthcare, or retail, which may have different characteristics (e.g., higher noise, non-stationarity, lack of strong periodicity), remains unverified.
4. The paper states that the SCWM models inter-channel dependencies on the frequency representations of the series. The mechanism for this is applying an MLP across the channel dimension for each frequency component. The intuition for why this is an effective way to model channel dependencies is not well explained. For example, does this allow the model to learn that a high value in frequency k for channel A should correlate with a low value in frequency k for channel B? A more intuitive explanation would be helpful.

[1] Shiyu Wang, Jiawei Li, Xiaoming Shi, Zhou Ye, Baichuan Mo, Wenze Lin, Shengtong Ju, Zhixuan Chu, Ming Jin: TimeMixer++: A General Time Series Pattern Machine for Universal Predictive Analysis. ICLR 2025
[2] Shiyu Wang, Haixu Wu, Xiaoming Shi, Tengge Hu, Huakun Luo, Lintao Ma, James Y. Zhang, Jun Zhou: TimeMixer: Decomposable Multiscale Mixing for Time Series Forecasting. ICLR 2024
[3] Lu Han, Xu-Yang Chen, Han-Jia Ye, De-Chuan Zhan: SOFTS: Efficient Multivariate Time Series Forecasting with Series-Core Fusion. NeurIPS 2024

**Questions:**

1. How does the softmax operation contribute? Can it be replaced by another normalization operation?
2. Are there any other constraints that can achieve similar performance? like L2- ball constrain?
3. Relationship to instance normalization or other normalization method?

---

### Official Review · Reviewer_VGLj · 2025-11-01

**Soundness:** 3
**Presentation:** 3
**Contribution:** 3
**Rating:** 6
**Confidence:** 3

**Summary:**

This paper studies overfitting in channel wise MLPs for multivariate time series forecasting and traces a key failure mode to extreme values (outliers). Building on Rademacher complexity arguments, the authors propose Simplex MLP, which constrains each channel mixing weight vector to the standard simplex. They embed this into a frequency–time pipeline dubbed FSMLP, comprising a Simplex Channel Wise MLP (SCWM) and a Frequency Temporal MLP (FTM). The loss combines time domain MSE and frequency domain MAE. On seven benchmarks , FSMLP shows strong accuracy and efficiency and also improves existing models when Simplex MLP is used as a drop in layer.

**Strengths:**

1.A concise and general mechanism.The paper links outliers to weight-norm inflation and mitigates overfitting to data spikes by constraining channel-mixing weights within the simplex. This approach is easy to implement, interpretable as a convex combination across channels, and theoretically justified by showing that the simplex constraint yields a tighter Rademacher bound.

2.Broad and scalable empirical evidence.FSMLP remains competitive or achieves state-of-the-art results on long-horizon forecasting tasks while maintaining high efficiency (overall complexity O(NL)). It also provides plug-and-play performance gains when integrated into existing models such as TSMixer and Autoformer.

**Weaknesses:**

1.Weak geometric motivation and interpretation.Although the paper describes the weight constraint as a “geometric constraint” that restricts parameters to lie within a standard simplex, its discussion of geometry remains largely formal, focusing only on the non-negativity and sum-to-one convex-combination conditions. The authors do not further explore the meaning of this geometric structure for the overall method, nor do they provide visualization or geometric interpretation. For instance, in classical linear programming, the simplex has a clear geometric implication—optimal solutions often occur at the vertices—yet no similar insight is discussed here. As a result, the “simplex constraint” in its current form appears more like an algebraic normalization operation than a genuinely geometry-inspired modeling approach, giving the impression that geometry serves merely as an external shell. A deeper exploration of the simplex’s geometric significance in future work would make the constraint more theoretically grounded and interpretable.

2.Simplistic constraint assumption.By restricting the channel-mixing weights to the standard simplex, the method effectively prevents weight inflation caused by extreme values, but it implicitly assumes that all inter-channel relationships are non-negative convex combinations—channels can only act in an additive, cooperative manner. The paper does not discuss the possibility of allowing negative weights, which in many multivariate time-series contexts play an inhibitory role: an increase in one channel may suppress signals in another, a phenomenon commonly observed in meteorological, economic, or sensor-network data. Completely excluding such signed interactions makes it difficult for the model to capture antagonistic or counteracting relationships explicitly. Consequently, the simplex constraint currently functions more as a numerical regularization to prevent overfitting rather than as a modeling hypothesis that fully reflects the nature of inter-channel interactions. Future work could consider introducing signed or zero-mean variants of the constraint to represent both positive and negative correlations—thereby capturing the inhibitory effects between channels while maintaining stability.

**Questions:**

1.On the geometric motivation.Could the authors elaborate on the geometric interpretation of the simplex constraint beyond non-negativity and normalization? For instance, does the geometry of the simplex  have any implication for optimization dynamics or model behavior? A visualization or geometric analysis would help clarify whether the constraint brings benefits beyond simple normalization.

2.On allowing signed or zero-mean weights.Have the authors considered extending the simplex constraint to allow signed or zero-mean weights? In many multivariate time-series settings, negative channel interactions represent inhibitory effects—an increase in one channel suppresses another. How would the proposed framework behave if such antagonistic relationships were permitted? Could a “centered simplex” or signed variant of the constraint preserve stability while improving modeling flexibility?

---

### Official Review · Reviewer_iSan · 2025-11-01

**Soundness:** 4
**Presentation:** 3
**Contribution:** 4
**Rating:** 6
**Confidence:** 5

**Summary:**

This paper introduces FSMLP, a novel model for time series forecasting that explicitly constrains MLP weights to a standard simplex (Simplex-MLP) to mitigate overfitting due to extreme values in multivariate time series data. The authors provide a theoretical justification using Rademacher complexity, showing that the proposed weight constraint reduces the model’s capacity to overfit noise. The approach is integrated into a frequency-domain framework (FSMLP) that includes Simplex Channel-Wise MLP (SCWM) and Frequency Temporal MLP (FTM) blocks. Extensive experiments on seven benchmark datasets and ablation studies demonstrate improved forecast accuracy, generalization, and computational efficiency over recent state-of-the-art baselines. The paper also shows that incorporating Simplex-MLP into existing models like TSMixer and Autoformer improves their performance.

**Strengths:**

- The paper gives a solid theoretical and empirical motivation for the overfitting problem associated with traditional channel-wise MLPs due to extreme values, as summarized in Table 1 and visually reinforced in Figure 1, which shows overfitting trend disparities among methods (FSMLP, TimesNet, TSMixer, Autoformer).
- The simplex constraint is rigorously justified with Rademacher complexity bounds (Section 5, Theorem 2), and a detailed proof is given in the Appendix, explaining why the constraint reduces generalization risk compared to L1/L2 regularization.
- The paper includes an exhaustive empirical comparison across diverse, standard benchmarks, demonstrating that FSMLP consistently outperforms prior models in both MSE and MAE for various input/output sequence lengths.
- The ablation study isolates the contributions of the simplex constraint, frequency transformation, and loss design. Figure 3 and Table 10 show the model achieves lower memory usage and faster inference/training times compared to prior models, demonstrating practical efficiency.
- The method is shown to generalize to and enhance other recent models, as evidenced in Table 9, where plugging Simplex-MLP into TSMixer/Autoformer yields notable performance gains across datasets.
- Mathematical derivations (e.g., equations for simplex projection, proofs of norm inflation in Section B.1/B.2) augment the presentation, demonstrating a solid grasp of both theoretical and practical considerations.

**Weaknesses:**

1. The math formulations and step-by-step derivations for projecting weights onto the simplex lack clarity around certain details, such as the computational complexity of each transformation and how they are implemented for large-scale matrices. The choices for $$f_\mathrm{trans}$$ (absolute, log, square) are described, but a more precise algorithmic statement or pseudocode for the entire weight update procedure, particularly for batch settings, is missing. This may hinder reproducibility and understanding of the exact workflow, especially in the context of PyTorch/TensorFlow weight updates.
2. The Rademacher complexity argument assumes bounded-norm, i.i.d. data. In real-world time series, substantial autocorrelation and non-stationarity violate these simplifications. The practical impact on generalization may be overestimated, and no sensitivity analysis explores cases of strongly autocorrelated or heavy-tailed sequences. A more thorough discussion or experiments on datasets with severe non-stationarity would solidify the contribution.
3. At several points, the paper argues that FSMLP is “universally” effective, yet the benchmarks are all from canonical, academic datasets. No evidence is given for extreme industrial settings (e.g., financial or medical data with even more skewed distributions), nor is the method tested on irregularly sampled or missing data. Similarly, statements like in Section 7 (“demonstrating scalability and robustness in large-scale, long-term forecasting tasks”) are somewhat overreaching given the size and scope of the benchmarks.

**Questions:**

1. Can the authors provide an explicit pseudocode or concrete algorithmic steps for the Simplex-MLP weight update (particularly the combination of $$f_{\text{trans}}$$ and $$f_{\text{norm}}$$ for batched tensors)? How efficient is this step computationally for high-dimensional weight matrices? Does it introduce runtime bottlenecks?
2. How robust is the FSMLP approach to highly non-stationary or autocorrelated time series, given the theoretical claims depend on i.i.d./boundedness assumptions? Do the authors see consistent improvements on, for example, financial or clinical datasets with heavy tails or strong patterns? If not, what failure modes emerge?
4. For scenarios requiring sparse or highly selective channel influence, does the simplex constraint risk underfitting? Would a convex-combination constraint (with temperature or sparsity control) provide a better regularization/expressivity tradeoff?

---

### Official Review · Reviewer_UooN · 2025-11-02

**Soundness:** 2
**Presentation:** 2
**Contribution:** 2
**Rating:** 2
**Confidence:** 4

**Summary:**

Overall, the paper’s motivation is unclear and, in some parts, potentially questionable.
The design rationale behind the proposed model architecture is not sufficiently articulated.
Several statements in the manuscript appear to be inaccurate or misleading, particularly in the Related Work section and the theoretical proofs.
In addition, the chosen baselines are outdated and do not include recent models from NeurIPS 2024, ICLR 2025, or ICML 2025. Given these issues, I believe the paper is not yet ready for publication and therefore recommend rejection.

**Strengths:**

1. The idea of introducing a simplex constraint on MLP weights is conceptually interesting.

2. The experimental section covers multiple benchmark datasets, providing a broad empirical context for evaluation.

**Weaknesses:**

1. In Table 1, it is unclear what type of “extreme values” the authors are referring to. The meaning of this term is ambiguous and requires clarification.

2. The content in Section 2.1 Time Series Forecasting, is not closely connected to the main topic discussed in this paper.

3. Regarding the description of the related work FreTS, the statements in lines 140–143 are incorrect. Furthermore, even on the authors’ own terms, the position expressed in lines 140–143 appears inconsistent with that in line 161.

4. In Figure 2,

    (i) The outer block is labeled “SCWM,” and the inner layer within the same block is also named “SCWM.” This creates ambiguity, as it is unclear whether “SCWM” refers to the entire module or a specific layer inside it.

    (ii) The two dashed diagonal lines should originate from the position of the SCWM layer.

    (iii) The *Simplex Constrain*, which is claimed as one of the main contributions of the paper, is not clearly illustrated or elaborated in the figure.

5. Issues in the Proof of Theorem 1 (Weight Norm Growth with Extreme Values)

    (i)  Incorrect use of spectral lower bound.   The proof employs $\lambda_{\min}(X^{\top}X)$ to lower-bound terms involving $\Delta X$, even though $\Delta X$ is an independent perturbation matrix unrelated to $X$.
   This substitution has no theoretical justification: one cannot assume that the smallest eigenvalue of $X^{\top}X$ constrains the behavior of $\Delta X$.
   A valid bound should depend on $\sigma_{\min}(\Delta X)$, the smallest singular value of the perturbation itself.

    (ii) Invalid inequality for difference of norms. The argument that $||\Delta XW - \Delta Y||_F^2 \ge \tfrac12||\Delta XW||_F^2 - ||\Delta Y||_F^2$ is mathematically incorrect.
   The standard triangle inequality only implies $||A-B||_F \ge |\ ||A||_F - ||B||_F|$;
   squaring this relation does **not** yield the linear form used in the proof.
   Consequently, the claimed lower bound on $||\Delta XW - \Delta Y||_F^2$ is unjustified.

6. Moreover, the proof of Theorem 2 (Rademacher Complexity of Simplex-MLP) contains errors in the treatment of the maximization domain and omits the necessary dimensional factor(s).

7. The numbering of the theorems is inconsistent: Theorem 1 (Weight Norm Growth with Extreme Values) in the main text becomes Theorem 3 in the appendix; similarly, Theorem 2 (Rademacher Complexity of Simplex-MLP) in the main text appears to correspond to another theorem also labeled Theorem 3 in the appendix.

8. Regarding the experimental section:

    (i) The selected baselines are relatively old and do not include more recent models published in ICLR 2025 or ICML 2025, which limits the competitiveness and relevance of the comparison.

    (ii) It would be very informative to visualize the learned weight matrix of Simplex-MLP to show how the simplex constraint affects the weight distribution.

    (iii)  Some tables repeat the same experimental results (for example, Table 3 vs. Table 9, and Table 5 vs. Table 8), which is unnecessary and could be consolidated.

    (iv) The paper lacks visualizations comparing predicted and ground-truth curves.

**Questions:**

1. Figure 1 presents experiments conducted on the ETTh1 dataset. The meanings of the horizontal and vertical axes are the same in both the left and right subfigures, but the curves differ. What causes this discrepancy? In addition, the figure’s intended message is unclear, as neither the caption nor the introduction provides a sufficiently detailed or insightful analysis.

2. Line 097: “and each coordinate is greater than or equal to zero.” Is this constraint applicable to real-world datasets? My concern is that when coordinate values are negative, such cases are ignored, even though negative values may naturally occur, for example, when variables exhibit negative correlations.

3. In Lines 235–247, the authors state that *“the operator $f_{\text{trans}}$ can be realized with each of the three following functions.”* I would like to know how one should determine which function to choose in different scenarios. In addition to these three functions, are there any other possible implementations of the operator $f_{\text{trans}}$ ? Furthermore, what do you consider to be the essential properties or constraints that $f_{\text{trans}}$  must satisfy?

4. In line 274, the phrase “N SCWM blocks and N FTM blocks” appears. Does *N* here refer to the number of series (as mentioned in “N series” in line 195)? It seems not.

---

### Meta-Review · Area_Chair_DMAc · 2025-12-20

**Summary:**

Although the paper introduces a simplex-constrained MLP and reports results on standard benchmarks, its core motivation is not clearly justified. Multiple reviewers questioned the correctness and rigor of the theoretical analysis. The experimental evaluation was also considered incomplete due to limited baseline coverage and dataset diversity.

**Reviewer Concerns:**

There was no rebuttal from the authors, and all major reviewer concerns remain outstanding.

**Reviewer Scores:**

Without a rebuttal, it is unlikely that any reviewer would increase their score. Reviewer UooN would clearly maintain the rejection recommendation. The higher-scoring reviewers, including iSan and VGLj, might reassess the severity of the unresolved theoretical and experimental issues and could lower their scores accordingly. Reviewer oDSf would also be unlikely to raise their score. Overall, the final score distribution would likely shift downward and remain insufficient to support acceptance.

---

### Decision · Program_Chairs · 2026-01-26

Reject